# Interpretable and Personalized Apprenticeship Scheduling: Learning Interpretable Scheduling Policies from Heterogeneous User Demonstrations

**Rohan Paleja, Andrew Silva, Letian Chen, and Matthew Gombolay**
Georgia Institute of Technology
Atlanta, GA 30332
{rohan.paleja, andrew.silva, letian.chen, matthew.gombolay}@gatech.edu

## Abstract

Resource scheduling and coordination is an NP-hard optimization requiring an efficient allocation of agents to a set of tasks with upper- and lower bound temporal and resource constraints. Due to the large-scale and dynamic nature of resource coordination in hospitals and factories, human domain experts manually plan and adjust schedules on the fly. To perform this job, domain experts leverage heterogeneous strategies and rules-of-thumb honed over years of apprenticeship. What is critically needed is the ability to extract this domain knowledge in a heterogeneous and interpretable apprenticeship learning framework to scale beyond the power of a single human expert, a necessity in safety-critical domains. We propose a personalized and interpretable apprenticeship scheduling algorithm that infers an interpretable representation of all human task demonstrators by extracting decision-making criteria via an inferred, personalized embedding non-parametric in the number of demonstrator types. We achieve near-perfect LfD accuracy in synthetic domains and 88.22% accuracy on a planning domain with real-world data, outperforming baselines. Finally, our user study showed our methodology produces more interpretable and easier-to-use models than neural networks ($p < 0.05$).

## 1   Introduction

Coordinating resources in time and space is a challenging and costly problem worldwide, affecting everything from the medical supplies we need to fight pandemics to the food on our tables. The manufacturing and healthcare industries account for a total of \$35 trillion [18] and \$8.1 trillion [28] USD, respectively. In manufacturing, scheduling workers – whether they be humans or robots – to complete a set of tasks in a shared space with upper- and lower-bound temporal constraints (i.e., deadline and wait constraints) is an NP-hard optimization problem [5], typically approaching computational intractability for real-world problems of interest.

Human domain experts efficiently, if sub-optimally, solve these NP-hard problems to coordinate resources using heterogeneous rules-of-thumb and strategies honed over decades of apprenticeship, creating unique heuristics depending on experts' varied experiences and personal preferences [23, 41]. Each expert has her own strategies, and it is common for factories and hospital wards to be run completely differently – yet effectively – across different shifts based upon the person in charge of coordinating the workers' activities [27, 33, 38]. The challenge we pose is to develop new *apprenticeship learning* techniques for capturing these heterogeneous rules-of-thumb in order to scale beyond the power of a single expert. However, such heterogeneity is not readily handled by traditional apprenticeship learning approaches that assume demonstrator homogeneity. A canonical example of this limitation is of human drivers teaching an autonomous car to pass a slower-moving car, where some drivers prefer to pass on the left and others on the right. Apprenticeship learning

approaches assuming homogeneous demonstrations either fit the mean (i.e., driving straight into the car ahead of you) or fit a single mode (i.e., only pass to the left).

The field of apprenticeship learning has recently begun working to relax the assumption of homogeneous demonstrations by explicitly capturing modes in heterogeneous human demonstrations [6, 7, 21, 26, 43]. One such approach, InfoGAIL [26], uses a generative adversarial setting with variational inference to learn discrete, latent codes to describe multi-modal decision-making. However, InfoGAIL requires access to an environment simulator, relies on a ground-truth reward signal, and is ill-suited to reasoning about resource scheduling and optimization problems, as we show in Section 5. Further, modern imitation learning frameworks lack interpretability, hampering adoption in safety-critical and legally-regulated domains [2, 10, 25, 44].

**Contributions** – Overcoming these key limitations of prior work, we develop a novel, data-efficient apprenticeship learning framework for learning from heterogeneous scheduling demonstration. The key to our approach is a neural network architecture that serves as a function approximator specifically designed for sparsity to afford easy "discretization" into a Boolean decision tree after training as well as the ability to leverage variational inference to tease out each demonstrator's unique decision-making criteria. In Section 5, we empirically validate that our approach, "Personalized Neural Trees", outperforms baselines even after discretization into a decision tree. Our contributions are as follows:

1. Formulate a personalized and interpretable apprenticeship scheduling framework for heterogeneous LfD that outperforms prior state-of-the-art approaches on both synthetic and real-world data across several domains ($+51\%$ and $+11\%$, respectively) through the use of personalized embeddings without constraining the number of demonstrator types.
2. Develop a methodology for converting a personalized neural tree into an interpretable representation that directly translates decision-making behavior. Our discretized trees also outperform previous benchmarks on synthetic and real-world data across several domains.
3. Conduct a user study that shows our post-processed interpretable trees are more interpretable ($p < 0.05$), easier to simulate ($p < 0.01$), and quicker to validate ($p < 0.01$) than their black box neural network counterparts.

## 2   Background

**Preliminaries** – LfD mechanisms are often based on a Markov Decision Process (MDP), a five-tuple $M = \langle S, A, T, \gamma, R \rangle$ where $S$ is a set of states, $A$ is a set of actions, $T : S \times A \times S \rightarrow [0, 1]$ is a transition function, in which $T(s, a, s')$ is the probability of being in state $s'$ after executing action $a$ in state $s$, $R: S \rightarrow \mathbb{R}$ (or $R : S \times A \rightarrow \mathbb{R}$) is the reward function, and $\gamma \in [0, 1]$ is the discount factor. The goal in LfD is to receive both a set of trajectories provided by a human demonstrator $\{\langle s_t, a_t \rangle, \forall t \in \{1, 2, \ldots T\}\}$ as well as an MDP\$R$, and then to recover a policy that can predict the correct state-action sequence a human would take in a novel situation.

**Imitation Learning** – There has been growing interest in tackling decision-maker heterogeneity [13, 17, 21, 26, 29, 32, 43]. Nikolaidis et al. [29] first used an expectation-maximization formulation to cluster decision-maker behavior before applying inverse reinforcement learning (IRL) for each cluster $k$. Negatively, this approach requires interaction with an environment model, and the IRL algorithm only has access to $\sim 1/k^{th}$ of the data to learn from. More recently, Li et al. [26] presented InfoGAIL, which used mutual information maximization to learn discrete, latent codes; however, InfoGAIL requires access to an environment simulator and a ground-truth reward signal. While InfoGAIL argues its latent codes afford interpretability, but its model structure is still a black-box neural network [36]. Hsiao et al. [21] presented an approach to discover latent factors within demonstrations using a categorical latent variable with limited expressivity. Finally, Tamar et al. [43] used a sampling-based approach to learn the modalities within the data, but the approach requires voluminous data due to the algorithm's high-variance estimation framework.

Researchers have also sought to learn scheduling policies from demonstration [1, 14, 15, 16]. Gombolay et al. [14] considers learning scheduling policies but does not consider heterogeneity. Berry et al. [1] proposed PTIME to learn to schedule calendar appointments; however, this approach requires manually soliciting user ranking data and computationally-intensive nonlinear optimization.

**Differentiable Decision Trees** – To both enable model interpretability and because prior work has shown the power of decision trees for apprenticeship scheduling [14], we seek to harness a decision tree architecture that is amenable to learning latent embeddings capturing heterogeneity. As such, we

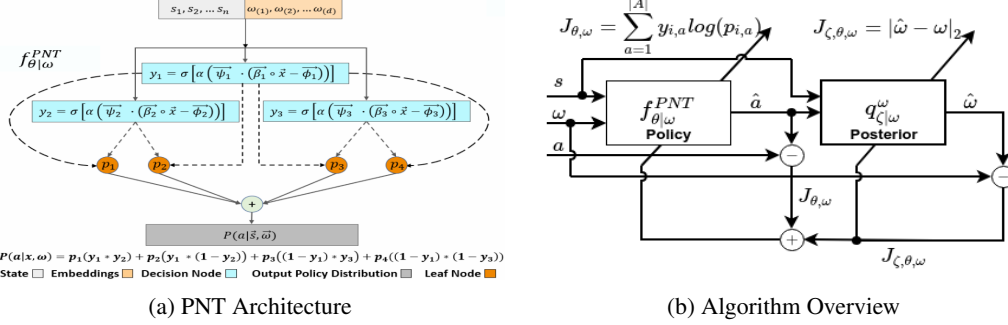

(a) PNT Architecture        (b) Algorithm Overview

Figure 1: The PNT architecture (left) displaying decision nodes, $y_i$, with evaluation equations, leaf nodes, $k$, with respective weights $p_k$, and output equation describing the calculation of the action PMF. An overview of our training algorithm (right) displaying the input/output flow of the policy and the posterior alongside their respective update equations.

build upon prior work in differentiable decision trees (DDTs) [40, 42] – a neural network architecture that takes the topology of a decision tree. Each decision node, $i$, is represented by a sigmoid activation function, $(1 + e^{-\alpha(\vec{\beta_i}\vec{s} - \phi_i)})^{-1}$, where the features vectors describing the current state, $\vec{s}$, are weighted by $\beta_i$, and a splitting criterion, $\phi_i$, is subtracted to form the splitting rule. $\alpha$ governs the steepness of the activation, where $\alpha \to \infty$ results in a step function. As we are learning a policy via apprenticeship learning, we adopt the formulation of [39] for the leaf nodes; i.e., each leaf node, $k$, is represented by a probability distribution over discrete actions, denoted $p_k$. We describe in Section 3.1 how we extend this formulation of a DDT for our framework.

# 3 Personalized and Interpretable Neural Trees

In this section, we present our apprenticeship framework that utilizes person-specific (personalized) embeddings, learned through backpropagation, which enables the apprenticeship learner to automatically adapt to a person's unique characteristics while simultaneously leveraging any homogeneity that exists within the data (e.g., uniform adherence to hard constraints).

## 3.1 Algorithm Overview

To effectively learn from heterogeneous decision-makers, we must capture the homo- and heterogeneity in their demonstrations, allowing us to learn a general behavior model accompanied by personalized embeddings that fit distinct behavior modalities. We contribute a novel apprenticeship learning model for resource scheduling, "Personalized Neural Trees", by extending DDTs in four important ways: 1) Personalized embeddings, $\omega$, as a latent variable representing person-specific modality (Section 3.2), 2) Variational inference mechanism to maximize the mutual information between the embedding and the modeled decision-maker (Section 3.2.1), 3) Counterfactual reasoning to increase data-efficiency (Section 3.2.2), and 4) Novel feature selector, $\vec{\psi_i}$, for each decision node to enhance interpretability (Section 3.3).

A Personalized Neural Tree (PNT) learns a model, $f_{\theta|\omega}^{PNT} : S \times \Omega \to [0, 1]^{|A|}$, of human demonstrator decision-making policies, where $\theta \in \Theta$ are the policy weights and $\omega \in \Omega$ ($\Omega \subset \mathbb{R}^d$) is the demonstrator-specific personalized embedding of length $d$, which is a tunable hyperparameter. Here, $\Omega = \{\omega_1, \omega_2, \ldots, \omega_P\}$ represents the set of all demonstrator personalized embeddings. The person-specific features $\omega$ identify the latent pattern of thinking for the current decision-maker. We note that the policy weights, $\theta \in \Theta$, are specifically defined as $\Theta = \alpha \times \Psi \times B \times \Phi \times \mathcal{P}$, where $\alpha$, $B$, and $\Phi$ are the parameters of the decision nodes (Section 3.2), $\mathcal{P}$ are the leaf parameters (Section 3.2), and $\Psi$ are a new set of parameters we introduce in Section 3.3 to enhance interpretability during post-training discretization.

Alongside learning a demonstrator's decision-making policy $f_{\theta|\omega}^{PNT}$, we introduce (Section 3.2.1) an information theoretic regularization model, similar to Chen et al. [8], to maximize mutual information between latent embeddings, $\omega$, and trajectories, $\tau$, by learning a model, $q_{\zeta|\theta}^{\omega} : S \times [0, 1]^{|A|} \to \mathcal{N}_{\Omega}$

represented by a neural tree (PNT\$\omega$) with weights $\zeta$, that approximates the true posterior $P(\omega|\tau)$. This induces the latent personalized embeddings to capture modality within demonstrator trajectories.

## 3.2 Personalized Neural Tree

We present architecture of $f_{\theta|\omega}^{PNT}$ as shown in Fig. 1a. First, a demonstrator-specific embedding (represented by $\omega \in \Omega$) is concatenated with state $\vec{s} \in S$ and routed directly to each decision node as $\vec{x} = [\vec{s}, \vec{\omega}]$). Each decision node in the PNT is conditioned on three differentiable parameters: weights $\vec{\beta} \in B$, comparison values $\vec{\phi} \in \Phi \subset [0,1]^{n+d}$, and selective importance vectors $\vec{\psi} \in \Psi$. When input data $\vec{x}$ is passed to a decision node, $i$, the data is weighted by $\vec{\beta_i}$ and compared against $\vec{\phi_i}$ as shown in Equation 1, where $\circ$ is the Hadamard product. The PNT algorithm uses its selective importance vector $\vec{\psi_i} \in [0,1]^{|\vec{x}|}$ before weighting by $\alpha$ and passing through a sigmoid to decide which "rule" is the most helpful to apply for this node; $y_i$ is the probability of decision node $i$ evaluating to TRUE.

$$y_i = \sigma[\alpha(\vec{\psi_i} \cdot (\vec{\beta_i} \circ \vec{x} - \vec{\phi_i}))] \tag{1}$$

Section 3.3 describes how this extension of the original formulation [42] enhances interpretability.

Leaf nodes, $k$, in the PNT maintain a set of weights over each output class denoted $\vec{p_k} \in \mathcal{P}$. Each decision node, $i$, along a path from the root to a leaf (i.e., a branch) output probabilities, $y_i$, for each such decision node. The branch's probabilities are multiplied to produce a joint probability of reaching the leaf in that branch given state $\vec{s}$ and the current demonstrator embedding $\omega_p$. Each leaf, $k$ contains a probability mass function (PMF), $\vec{p_k}$, where $\vec{p_{k,a}}$ is the probability of applying action $a$ for the branch leading to leaf node, $k$. This probability distribution, $\vec{p_k}$, for leaf, $k$, is multiplied by its corresponding branches' joint probability. Finally, the products of all leaf vectors with their branch's joint probability is summed to produce the final network output, a PMF for actions given state, $s$, and embedding, $\omega_p$. An example is shown in Fig. 1a complete with an equation summarizing the output.

### 3.2.1 Maximizing Mutual Information

The parameters, $\zeta$, $\theta$, and $\omega$, are updated via a cross-entropy loss and mutual information maximization loss, as discussed in Section 3.4. Maximizing mutual information encourages $\omega$ to correlate with semantic features within the data distribution (i.e., mode discovery) [8, 26]. Yet, maximizing mutual information between the trajectories and latent code, $G(\omega; \tau)$, is intractable as it requires access to the true posterior, $P(\omega|\tau)$. Therefore, researchers employ the evidence lower bound (ELBO) of the mutual information $G(\omega; \tau)$, as shown in Equation 2. Maximizing $G(\omega; \tau)$ incentivizes the policy, $f_{\theta|\omega}^{PNT}$, to utilize the latent embedding $\omega$ as much as possible.

$$G(\omega; \tau) = H(\omega) - H(\omega|\tau)$$
$$\geq \mathbb{E}_{\omega_p \sim \mathcal{N}(\vec{\mu}_p, \vec{\sigma}_p^2), a \sim f_{\theta|\omega}^{PNT}}[log(q_{\zeta|\theta}^{\omega}(\omega_p|s_p^t, a_p^t))] + H(\omega) = L_G(f_{\theta|\omega}^{PNT}||q_{\zeta|\theta}^{\omega}) \tag{2}$$

In our approach, we make use of continuous personalized embeddings which allow for greater expressivity in the embedding space, $\Omega$. As such, we utilize a mean-squared error (MSE) loss between a sample from the approximate posterior (modeled as a normal distribution with constant variance) and the current embedding. A derivation of the equivalence between using the MSE and log-likelihood loss to maximize the posterior is attached in the supplementary material.

### 3.2.2 Counterfactual Reasoning

We further enhance our model's learning capability through counterfactual reasoning [3, 4, 11, 14, 22, 30, 31]. Based upon the insight in prior work in homogeneous apprenticeship scheduling [14] that counterfactual reasoning was critical for learning scheduling strategies from demonstration, we adopt counterfactual reasoning through pairwise comparisons. We present a novel extension to construct counterfactuals in Equations 3-4 that leverages person-specific embeddings as pointwise terms.

$$z_{a,a'}^{t,p} := [\omega_p, \bar{x}^t, x_a^t - x_{a'}^t], y_{a,a'}^t = 1 \text{ for } \forall a' \in A \setminus a \tag{3}$$

$$z_{a',a}^{t,p} := [\omega_p, \bar{x}^t, x_{a'}^t - x_a^t], y_{a',a}^t = 0 \text{ for } \forall a' \in A \setminus a \tag{4}$$

At each timestep, we observe the decision, $a$, that person, $p$, made at time $t$. From each observation, we then extract 1) the feature vector describing that action, $x_a^t$, from state $s^t$, 2) the corresponding

feature, $x^t_{a'}$, for an alternative action $\forall a' \in A \setminus a$, 3) a contextual feature vector capturing features common to all actions, $\bar{x}^t$, and 4) the person's embedding, $\omega_p$. We note that each demonstrator has their own embedding which is updated through backpropagation.

$$\hat{P}(a|t,p) \sim \sum_{a' \in A} f(a,a',p) \tag{5}$$

---

**Algorithm 1** PNT Training

**Input**: data $\vec{s} \in S$, labels $a \in A$, embeddings $\omega \in \Omega$
**Output**: $f^{PNT*}_{\theta|\omega}$

1: Initialize $f^{PNT}_{\theta|\omega}$, $q^{\omega}_{\zeta|\theta}$, $\Omega$
2: **for** i epochs **do**
3:      Sample data of person $p$ at time $t : \vec{s}^t_p, a^t_p$
4:      $\vec{x}^t_p \leftarrow [\omega^{(i)}_p, \vec{s}^t_p]$
5:      $\hat{a}^t_p \leftarrow f^{PNT}_{\theta|\omega}(\vec{x}^t_p)$
6:      $\mu_p, \sigma_p \leftarrow q^{\omega}_{\zeta|\theta}(\vec{s}^t_p, \hat{a}^t_p)$
7:      $\hat{\omega}^{(i)}_p \sim \mathcal{N}(\vec{\mu}_p, \vec{\sigma}_p{}^2)$
8:      $J_{\zeta,\theta,\omega} = |\hat{\omega}^{(i)}_p - \omega^{(i)}_p|$
9:      $J_{\theta,\omega} = CrossEntropy(\hat{a}^t_p || a^t_p)$
10:     $J \leftarrow J_{\theta,\omega} + J_{\zeta,\theta,\omega}$
11:     $[\omega_p, \theta, \zeta]^{(i+1)} \leftarrow [\omega_p, \theta, \zeta]^{(i)} - \eta \nabla_{\omega_p,\theta,\zeta} J$
12: **end for**

**Algorithm 2** PNT Run-time Adaptation

**Input**: data $\vec{s}^t_{p*} \in S$, $f^{PNT*}_{\theta|\omega}$, training embeddings mean $\bar{\Omega}$
**Output**: Demonstrator $p*$'s action: $\hat{a}^t_p$

1: $\omega^t_{p*} = \bar{\Omega}$
2: **for** t in range(1,T) **do**
3:      $\vec{x}^t_{p*} = [\omega^t_{p*}, \vec{s}^t_{p*}]$
4:      $\hat{a}^t_{p*} \leftarrow f^{PNT*}_{\theta|\omega}(\vec{x}^t_{p*})$
5:      $a^t_{p*} \leftarrow ObserveAction()$
6:      $J_{\theta,\omega} = CrossEntropy(\hat{a}^t_{p*} || a^t_{p*})$
7:      $\omega^{(t+1)}_p \leftarrow \omega^{(t)}_p - \eta \nabla_{\theta,\omega} J_{\theta,\omega}$
8: **end for**

---

Given this dataset, the apprentice is trained to output a pseudo-probability, $f(a, a', p)$ of action $a$ being taken over action $a'$ at time $t$ by the human decision-maker $p$ described by embedding $\omega_p$, using features $z^{t,p}_{a,a'}$. To predict the probability of taking action $a$ at timestep $t$, we marginalize over all other actions, as shown in Equation 5. Finally, the action prediction is the argument max of this probability, $\hat{a} = \arg \max_{a \in A} \hat{P}(a|t,p)$. We term models that use counterfactual reasoning as pairwise models.

> **Nota Bene**: *While counterfactuals have been exploited in prior work, ours is the first to our knowledge that incorporates variational inference for counterfactual learning from heterogeneous demonstration.*

## 3.3 Interpretability via Discretization

In our work, the PNT is able to learn over datasets from heterogeneous demonstrators with high performance while still being able to convert back into a simple, interpretable decision tree post-training. Our formulation, as shown in Equation 1, includes two important augmentations to the original DDT formulation to allow for discretization post-training: 1) The per-node feature selector vector, $\vec{\psi}_i$, that learns the relative importance of each candidate splitting rule, $(\beta_{i,j}x_{i,j} - \phi_{i,j})$ for each feature, $j$, and node, $i$, and 2) a per-feature splitting criterion, $\vec{\phi}_{i,j}$, that enables us to simultaneously curate per-node and per-feature splitting criteria.

During discretization of the PNT to its interpretable form, we apply the following operations to each decision node, $i$: 1) Set the argument max of $\vec{\psi}_i$ to 1 and all other elements to zero; 2) Set $\alpha \leftarrow \infty$. For each leaf node, $i'$, we likewise set the argument max of $\vec{p}_{i'}$ to one and all non-maximal elements zero. The result of these operations is that each decision node has a single, Boolean splitting rule as per a standard decision tree and each leaf node dictates a single action to be taken. This procedure produces a simple yet powerful decision tree, which we show in Section 5 outperforms all baselines even in its interpretable form.

## 3.4 Training and Runtime Procedure

**Offline –** At the start of training (Algorithm 1), each $\omega_p$ is a vector of uniform values. A state is sampled, $s^t_p$ at time $t$, for demonstrator $p$, as well as the person-specific embedding, $\omega^{(i)}_p$ at training iteration $i$, to produce a concatenated input, $\vec{x}^t_p$ as shown in lines 3 and 4. Policy $f^{PNT}_{\theta|\omega}$ uses input $\vec{x}^t_p$ to predict the demonstrator's action in that state, $\hat{a}^t_p$, as shown in line 5. The predicted action, $\hat{a}^t_p$, and state, $s^t_p$, are then used to recover a normal distribution, $\mathcal{N}(\vec{\mu}_p, \vec{\sigma}_p^2)$, representing that user's

personalized embedding $\omega_p^{(i)}$. By sampling from this distribution, $\hat{\omega}_p^{(i)} \sim \mathcal{N}(\vec{\mu_p}, \vec{\sigma_p}^2)$, we can estimate the accuracy of our approximate posterior by computing the difference between the current embedding and the sampled embedding shown in lines 6 and 7. The learning from demonstration loss is then computed as the cross entropy loss between the true action $a_p^t$ and the predicted action $\hat{a}_p^t$. Summed together, we have a total loss $J$ that is dependent on $\zeta$, $\theta$, and $\omega$, as shown in lines 7-10. This loss is then used to update model parameters $\theta$, personalized embedding $\omega$, and embedding regularization parameters $\zeta$ via SGD [34], as shown in line 11. This process is repeated until a convergence criterion is satisfied. An overview of this training procedure is displayed in Fig. 1b.

**Online –** When applying the algorithm during runtime for a new human demonstrator, $p'$, the model updates the embedding, $\omega_{p'}$; however, $\theta$ remain static. This online update utilizes the information provided after every timestep (i.e., the true action) to converge on the type of current demonstrator in embedding space. The personalized embedding $\omega_{p'}$ for a new human demonstrator is initialized to the mean of the embeddings of demonstrators in the training set and updated as we gain more information, as shown in Algorithm 2.

**Interpretable Policy –** Once this tuning process has finished, the person-specific policy can be converted into an interpretable tree, through discretization of our PNTs.

**Covariate Shift –** To remedy the co-variate shift typically encountered with policy-based apprentice-ship learning, DAgger [35] was proposed for problems where there is access to the environment. In Section 5, we show that pre-training with PNTs leads to a significant increase in performance for DAgger-based training while also reducing the number of environment samples DAgger requires.

## 4   Evaluation Environments

We utilize three environments to evaluate the utility of our personalized apprenticeship scheduling framework. Additional details about each domain are provided in the supplementary material.

**1) Synthetic Low-Dimensional Environment** The synthetic low-dimensional environment represents a simple domain where an expert will choose an action based on the state and one of two hidden heuristics. This domain captures the idea that we have homogeneity in conforming to constraints z and strategies or preferences (heterogeneity) in the form of $\lambda$. Demonstration trajectories are given in sets of 20 (which we denote a complete schedule), where each observation consists of $x^t \in \{0, 1\}$ and $z^t \in \mathcal{N}(0, 1)$, and the binary output is $y^t$. Exact specifications for the computation of the label are given by the observation of $y = x * \mathbb{1}_{(z>=0 \wedge \lambda=1) \vee (z<0 \wedge \lambda=2)}$, where $\mathbb{1}$ is the indicator function.

**2) Synthetic Scheduling Environment** The second environment we use is a synthetic environment that we can control, manipulate, and interpret to empirically validate the efficacy of our proposed method. For our investigation, we leverage a jobshop scheduling environment built on the **XD[ST-SR-TA]** scheduling domain defined by Korsah [24], representing one of the hardest scheduling problems. In this environment, two agents must work together to complete a set of 20 tasks that have upper- and lower-bound temporal constraints (i.e., deadline and wait constraints), proximity constraints, and travel-time constraints. Schedulers have a randomly-generated task-prioritization scheme dependent upon task deadline, distance, and index. The decision-maker must decide the optimal sequence of actions according to the decision-maker's own criteria. This domain is a more complex variant of the domain in Gombolay et al. [14] as we have demonstrations of heterogeneous scheduling strategies.

**3) Real-world Data: Taxi Domain** We evaluate our algorithm with actual human decision-making behavior collected in a variant of the Taxi Domain in [9]. This domain describes a **ND[ST-SR-TA]** scheduling domain as defined by Korsah [24]. Our environment has three locations: the village, the airport, and the city. The taxi driver has the objective of picking up a passenger from the city or village. A dataset of 70 human-collected tree policies to solve this task (given with leaf node actions such as "Drive to X" and "Wait for Passenger", and decision node criterion depending on the amount of wait time, traffic time, and current location) are used to generate heterogeneous trajectories.

Table 1: A comparison of heterogeneous LfD approaches. Our method achieves superior performance. Interpretable approaches are shown in the right-hand table.

| Method | Low-Dim | Scheduling | Taxi |
|---|---|---|---|
| **Our Method** | **97.30 ± 0.3%** | **96.13 ± 2.3%** | **88.22 ± 0.6%** |
| Sammut et al. | 55.36 ± 1.2% | 5.00 ± 0.0% | 76.16 ± 0.3% |
| Nikolaidis et al. | 54.23 ± 2.5% | 5.00 ± 0.0% | 76.16 ± 0.3% |
| Tamar et al. | 55.83 ± 0.6% | 9.78 ± 0.3% | 60.93 ± 2.8% |
| Hsiao et al. | 56.06 ± 1.1% | 11.25 ± 0.1% | 76.19 ± 0.4% |
| InfoGAIL | 54.66 ± 3.4% | 25.72 ± 5.5% | 75.51 ± 0.8% |
| DDT | 55.28 ± 1.8% | 52.35 ± 0.7% | 76.70 ± 0.7% |

| Method | Low-Dim | Scheduling | Taxi |
|---|---|---|---|
| **Our Method (Interpretable)** | **96.13 ± 2.5%** | **99.66 ± 0.5%** | 77.73 ± 1.9% |
| **Our Method (DT + $\omega$)** | 53.66 ± 2.4% | 32.85 ± 0.1% | **87.85 ± 0.5%** |
| Gombolay et al. | 55.76 ± 1.4% | 45.50 ± 2.0% | 75.88 ± 0.7% |
| Vanilla DT | 55.76 ± 1.4% | 32.4 ± 0.7% | 74.90 ± 0.2% |

## 5 Results and Discussion

We benchmark our approach against a variety of baselines [14, 21, 26, 29, 37, 43][2]. Accuracy is the $k$-fold cross-validation, multi-class, classification accuracy for state-action pairs.

**1) Synthetic Low-Dimensional Environment –** Table 1 shows that our method for learning a continuous, personalized embedding sets the state-of-the-art ($95.30\% \pm 0.3\%$) for solving this latent-variable classification problem. Even after discretizing to an interpretable form, our method is still able to outperform all baselines, achieving a $96.13\% \pm 2.49\%$ accuracy. A graphical depiction of the interpretable PNT model, generated through discretization is provided in the supplementary.

**2) Synthetic Scheduling Environment –** Table 1 shows that our personalized apprenticeship learning framework outperforms all other approaches, achieving $96.13\% \pm 2.3\%$ accuracy in predicting demonstrator actions before conversion to an interpretable policy. After discretizing to an interpretable form, our method is able to achieve $99.66\% \pm 0.5\%$ accuracy[1]. Furthermore, benchmarks that seek to handle heterogeneity [21, 26, 43] are unable to handle the complexity associated with resource coordination problems, with InfoGAIL achieving only $25.72\% \pm 5.5\%$ accuracy. We provide a sensitivity analysis for this domain within our supplementary material by considering noisy demonstrations and varying the amount of data available to train the algorithm.

**3) Real-world Data: Taxi Domain –** As seen in Table 1, our personalized apprenticeship learning framework outperforms all other benchmarks and is the only method to achieve over $80\%$. Even after discretizing to an interpretable form, our method again outperforms all baselines from prior work. We posit that all other methods overfit to the most prevalent behavior, and are unable to tease out the heterogeneity represented within the training dataset.

**Interpretability –** We validate the effectiveness of using the discretized PNT architecture versus several interpretable architectures in Table 1. In two out of three of our environments, using our discretized PNT architecture results in a large performance gain (42.57% in the low-dim and 38.01% in the scheduling environment). In one domain, our method for distilling a DT using our PNT architecture's learned embeddings appended to the states for training a DT policy was better.

**Understanding Performance of Baselines –** Each baseline has particular flaws that results in its low performance on scheduling problems. Sammut et al. [37] simply assumes homogeneity and serves as a lowerbound. Nikolaidis et al. [29] has two-step approach that first clusters to find modes and then trains policies; thus, there is no feedback symbol to adjust clustering-established modes. Tamar et al. [43] utilizes a sampling-based approach which is acknowledged to require larger data sets. Hsiao et al. [21] uses a categorical variable for modes results in limited expressively. InfoGAIL [26] does not have access to a ground-truth reward signal nor the environment. Our approach is the only method that both includes a decision tree-like architecture, helpful for apprenticeship scheduling [14], while also allowing for variational inference.

**Performance of Pretrained Policies with DAgger –** In deploying our pretrained policies within the scheduling environment, our metrics to verify whether our pre-trained PNT (PT-PNT) policies with DAgger are able to outperform randomly-initialized PNT (RI-PNT) policies with DAgger are 1) to maximize the number of tasks scheduled before a terminal state and 2) maximize the number of successful schedules. Our PT-PNT is trained on a set of 150 schedules and then with 500 episodes of DAgger. Our RI-PNT is given 650 episodes of DAgger. We find that our PT-PNT outperforms a RI-PNT by 28.57% in successful schedule completion and 12.5% in the number of tasks completed

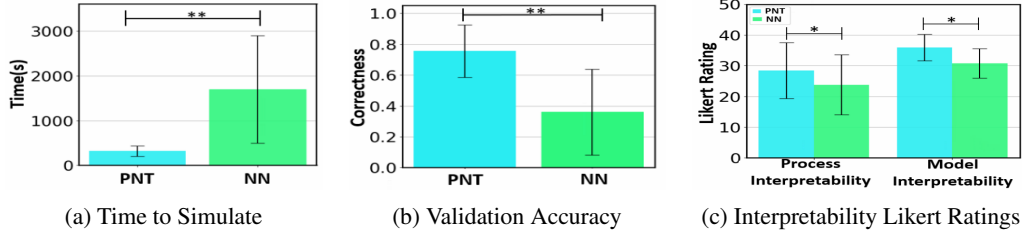

| (a) Time to Simulate | (b) Validation Accuracy | (c) Interpretability Likert Ratings |

Figure 2: The findings of our user study. We find significance for hypotheses **H1**, **H2**, and **H3**.

before failure. This result shows the benefits of pre-training using our framework and that our approach is amenable to DAgger-based training.

## 6 Interpretability User Study

Thus far, we have shown across a variety of datasets that our counterfactual PNT algorithm is able to achieve SOTA performance in learning from heterogeneous decision-makers. To show that our models are interpretable, we assess whether the counterfactual PNT is useful in the hands of end users. Accordingly, we conducted a novel user study to assess the interpretability of our framework. We design an online questionnaire that asks users to make predictions following each a counterfactual decision tree (PNT) and a neural network (NN). Detail about the generation of these models is in the supplementary material. We explore three hypotheses: tree-based decision-making models are more interpretable **(H1)**, quicker to validate **(H2)**, and are easier to simulate **(H3)** than neural networks. To test **(H1)**, we ask users to answer a 13-item Likert questionnaire assessing whether the user understands the components of the decision-making model (i.e., model interpretability) and how to translate an input to an output (i.e., process interpretability) after utilizing each decision-making framework. These subjective measurements provide a practical gauge of how interpretable the decision-making models are in the hands of end-users in a **XD[ST-SR-TA]** scheduling domain as defined by Korsah [24]. To test **(H2)** and **(H3)**, we record the time required for a user to compute the model's output given a set of inputs, and measure the user's ability to correctly determine the model's output given a set of inputs, respectively.

### 6.1 User Study Results and Discussion

Our IRB-approved anonymous survey was completed by twenty adult university students. Fig. 2 depicts the results testing **H1-H3**. The complete analysis is located in the supplementary material. **H1:** We test for normality and homoscedasticity and do not reject the null hypothesis in either case, using Shapiro-Wilk ($p > 0.3$ and $p > 0.7$) and Levene's Test ($p > 0.5$ and $p > 0.1$). We perform a paired t-test and find that tree-based models were rated statistically significantly higher than neural networks on users' Likert scale ratings for model interpretability and overall process interpretability ($p < 0.05$ and $p < 0.05$). **H2:** We perform a Wilcoxon signed-rank test on the per-model time to determine an output and find that tree-based models were statistically significantly quicker to validate than neural networks ($p < 0.01$). **H3:** We test for normality and homoscedasticity and do not reject the null hypothesis in either case, using Shapiro-Wilk ($p > 0.2$) and Levene's Test ($p > 0.4$). We perform a paired t-test and find that users using tree-based models statistically significantly achieved higher overall correctness scores compared to NN based models ($p < 0.01$), supporting **H3**. Given these positive results, we believe our model sets a new state-of-the-art in accuracy for heterogeneous LfD (Table 1) and also a strong step towards making such models more interpretable.

## 7 Conclusion

We present an apprenticeship scheduling framework for learning from heterogeneous demonstrators, leveraging a Personalized Neural Tree that is able to capture the homo- and heterogeneity in scheduling demonstrations through the use of personalized embeddings. The design of our PNT allows for translation into an interpretable form while maintaining a high level of accuracy. We demonstrate that our approach is notably superior to standard apprenticeship learning models and several approaches used in multi-modal behavior learning on synthetic and real-world data across three domains. Finally, we conduct a novel user study to assess the interpretability between our discretized trees and neural networks and find that our discrete trees are more interpretable ($p < 0.05$), easier to simulate ($p < 0.01$), and quicker to validate ($p < 0.01$).

# 8 Broader Impact

Our interpretable apprenticeship scheduling framework has broad impacts on society and the learning from demonstration community. Our interpretable trees can give key insight into the behavior of a machine-learning-based agent, allowing a human to verify that safety constraints are being met and increasing the trustworthiness of the autonomous agent. Furthermore, these trees allow human operators to follow the decision step-by-step, allowing for verification [12, 19], and holding machines accountable [20].

**Beneficiaries –** Our work has the potential to benefit all human-machine collaborations, providing improved transparency, and strengthening the trustworthiness of machine teammates through policy verification. Our research contributions additionally benefit research and laboratories pursuing learning from diverse human data (which commonly contains heterogeneity).

**Negatively affected parties –** With any model, we believe it is important to gather consent before utilizing one's data. As our model can be used by humans as both a forward model to understand decision-maker behavior and as an inverse model to infer modality, there is a possibility of discovering latent characteristics about individuals that may reflect negatively upon them.

**Implications of failure –** Failure of our approach to produce high-accuracy behavior during deployment will result in a lack of trust towards the system. In the worst case, careless application may contribute to misunderstandings causing damage from a deployed robot.

**Bias and Fairness –** The learned behavior of our PNTs will be biased towards demonstrators within the training set. If the collected set excludes certain persona, the behavior of these persona will not be represented by our PNT. However, it should be noted that as our approach is able to better take into account heterogeneity within the training data compared to other apprenticeship learning approaches. In other words, our framework is better able to represent the entire population rather than overfitting to the most prevalent demonstrator behavior than previous approaches.

**Impact on LfD community –** Personalized Neural Trees can easily be extended to a variety of domains, increasing the data-efficiency, accuracy, and utility of learning-from-demonstration with multiple human demonstrators. We demonstrate this by using a PNT to learn kinesthetic robot table tennis demonstrations. We provide details about this domain, the collection process, and the results in the supplementary material.

**Reproducibility –** Following the NeurIPS Reproducability Checklist, we upload all code here. Within this repository, we provide collected real-world datasets, code to generate synthetic data, and code to run all benchmarks. Alongside this, we attach trained models for each domain. Further in the supplementary material, we provide specifications of our hyperparameters, descriptions of our computing infrastructure, and other details regarding runtime.

## Acknowledgments and Disclosure of Funding

This work was sponsored by MIT Lincoln Laboratory grant 7000437192, the Office of Naval Research under grant N00014-19-1-2076, NASA Early Career Fellowship grant 80HQTR19NOA01-19ECF-B1, and a gift to the Georgia Tech Foundation from Konica Minolta, Inc.

## Footnotes

[1]To infer the embeddings for the interpretable form of our PNT model, we utilize a pre-discretized version of the PNT to learn a demonstrator's embeddings, which is run prior or concurrently with the discretized version.

[2]An offline version of InfoGAIL [26] is used, as access to a simulator and ground truth reward signal, $R$, is not available in many real-world domains.

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
