[Supplementary Material 1]



**Introduction Standard**

Hello,

Thank you for agreeing to do this survey!

For this survey, you will be asked to use 2 different types of decision-making frameworks and rate them based on their interpretability.

I will first provide you with instructions on how to use a certain type of decision-making framework. After this, I will ask you to click to the "next" button and use the decision-making framework to trace an input to an output. **I will ask that you do not take breaks on this page, as the speed at which you complete this exercise is timed**. At this point, you will be asked to answer some subjective questions about the decision-making framework used and the overall process of choosing a decision given the method.

This process will be repeated two times for two different decision-making frameworks. This should take you approximately 15 minutes.

Please have a paper, writing utensil, and calculator at hand as you may need keep track of up to 10 numbers at once.  You may use a code editor as a calculator but please do not use it to assist with matrix multiplication. **As compensation, you will receive a $5 Amazon gift card.** Note that you must meet a minimum precalculated survey statistic to receive the compensation.

This survey has been approved by the Georgia Institute of Technology IRB. We recommend that you conduct this survey on your computer rather

than on a mobile device.

**Standard Decision Tree Introduction**

A standard decision tree is a decision-making model in which each internal node represents a "test" on an attribute (e.g. whether you have completed your homework), each branch represents the outcome of the test, and each leaf node represents a class label (e.g., you can go play outside). The paths from root to leaf represent classification rules.

In our trees, moving LEFT in the decision tree is associated a "test" of an attribute being true.

Let's do a quick example.

Standard Decision Tree representing the decision of "Should I write a blog post?".

In the decision tree above, the input into the decision tree would [Do I understand the concept?, Do I feel tired?, Do I have tea nearby?] and the outputs are either yes or no.

For example,

If we received the input of [yes, yes, no], the output would be No.

**On the next page, you will be given a decision tree representing scheduling behavior. Given input data about the difficulty of three tasks and whether the scheduler prefers to handle difficult tasks or easy tasks, you will have to use the decision tree given to decide which task to schedule.**

**The input array will be of size 5, and the decision tree will reference the corresponding element using {#} notation.**

**For example, for the input data of [11, 22, 33, 44, 55], {3} refers to 33.**

**Standard Decision Tree Test**

**These page timer metrics will not be displayed to the recipient.**
#EditSection, TimingFirstClick#: *0 seconds*
#EditSection, TimingLastClick#: *0 seconds*
#EditSection, TimingPageSubmit#: *0 seconds*
#EditSection, TimingClickCount#: *0 clicks*

## Input data:

## [2, 1, 3, 0.40, 0.60]

Standard decision tree representing scheduling behavior.

**Standard Decision Tree**

What is the output of this decision tree given the input above? Please type the number in the subscript.

For example, if the output you receive is tau_25, you would type 25.

---

**Standard Decision Tree Survey**

Please fill out the questions below. The phrase "decision-making model" refers specifically to the graphic on the previous page. The phrase overall "decision-making process" refers to the entire process starting from being given input(s) to answering the question(s).

| | Very Strongly Disagree | Strongly Disagree | Disagree | Neutral | Agree | Strongly Agree | Very Strongly Agree |
|---|---|---|---|---|---|---|---|
| The decision-making model is interpretable. | ○ | ○ | ○ | ○ | ○ | ○ | ○ |

| | Very Strongly Disagree | Strongly Disagree | Disagree | Neutral | Agree | Strongly Agree | Very Strongly Agree |
|---|---|---|---|---|---|---|---|
| I understand the behavior represented within the decision-making model. | ○ | ○ | ○ | ○ | ○ | ○ | ○ |
| The decision-making model logic is easy to follow. | ○ | ○ | ○ | ○ | ○ | ○ | ○ |
| The decision-making model does not make sense. | ○ | ○ | ○ | ○ | ○ | ○ | ○ |
| The decision-making model is difficult to understand. | ○ | ○ | ○ | ○ | ○ | ○ | ○ |
| I could follow the rules of this decision-making model with ease. | ○ | ○ | ○ | ○ | ○ | ○ | ○ |
| I like the level of readability of this decision-making model. | ○ | ○ | ○ | ○ | ○ | ○ | ○ |
| The overall decision-making process is easy to comprehend. | ○ | ○ | ○ | ○ | ○ | ○ | ○ |
| I understand the overall process of choosing an output given input(s). | ○ | ○ | ○ | ○ | ○ | ○ | ○ |
| This overall decision-making process logic is easy to follow. | ○ | ○ | ○ | ○ | ○ | ○ | ○ |
| The overall decision-making process does not make sense. | ○ | ○ | ○ | ○ | ○ | ○ | ○ |
| The overall decision-making process is difficult to understand. | ○ | ○ | ○ | ○ | ○ | ○ | ○ |
| I could follow the rules of this decision-making tool with ease. | ○ | ○ | ○ | ○ | ○ | ○ | ○ |

## Standard Neural Network Introduction

A standard neural network is decision-making model in which a set of inputs (e.g., hours of homework per week) are transformed through multiplication and activation functions to produce an output (e.g., will you ace your math test?).

Let's do a quick example.

Neural network example

The above neural network has 3 input nodes (represented by the 3 nodes in the left-most of the diagram), 3 hidden nodes (represented by the 3 nodes in the middle of the diagram and 2 output nodes.

For our example, the 3 input nodes (features) refer to [how many hours of homework you do a day, how many video games you play a day, and how many hours you read a day] and the 2 output nodes refer to the likelihood of receiving a failing grade on your math test, and the likelihood of receiving a passing mark on your math test.

Say our input features are [1,0,2] corresponding to 1 hours of homework a day, 0 video games, and 2 hours of reading.

## Relu Function

$$f(x) = \begin{cases} 0 & \text{for } x < 0 \\ x & \text{for } x \Rightarrow 0 \end{cases}$$

## Softmax Function

$$\mathrm{Softmax}(x_i) = \frac{\exp(x_i)}{\sum_j \exp(x_j)}$$

## Weights of the neural network

$$W_1 = \begin{array}{c} \\ h1 \\ h2 \\ h3 \end{array} \begin{array}{ccc} i1 & i2 & i3 \\ \begin{bmatrix} 1 & 0 & 0 \\ 2 & 1 & 1 \\ 3 & 0 & 2 \end{bmatrix} \end{array}$$

$$B_1 = \begin{array}{c} b1 \\ b2 \\ b3 \end{array} \begin{bmatrix} 1 \\ 0 \\ 0 \end{bmatrix}$$

$$W_2 = \begin{array}{c} \\ o1 \\ o2 \end{array} \begin{array}{ccc} h1 & h2 & h3 \\ \begin{bmatrix} 0 & 0 & 0 \\ 3 & 1 & 2 \end{bmatrix} \end{array}$$

$$B_2 = \begin{array}{c} b1 \\ b2 \end{array} \begin{bmatrix} 0 \\ 1 \end{bmatrix}$$

The outputs o1 and o2 can be computed through the equation below.

H = W_1 * x + B_1

H = ReLU(H)
O = W_2 * H + B_2

Here, x is our input column vector [1,0, 2]
W_1, W_2, B_1, and B_2 are shown in the images.

The multiplication of H breaks down to
To get to value of h1, we do (1 * 1 + 0 * 0 + 2 * 0) + 1 = 2
To get to value of h2, we do (1 * 2 + 0 * 1 + 2 * 1) + 0 = 4
To get to value of h3, we do (1 * 3 + 0 * 2 + 2 * 2) + 0 = 7

The ReLU function is discussed above. For the vector H, it turns all negative numbers to 0, and leaves all positive the same. Thus, the ReLU function has no effect.

The computation of O breaks down to
To get to value of o1, we do (1 * 0 + 4 * 0 + 7 * 0) + 0 = 0
To get to value of o2, we do (1 * 3 + 4 * 1 + 7 * 2) + 1 = 22

To compute final probabilities, we will utilize the softmax function (shown above).
o1 = e^0 / (e^0+ e^22) = 2.78e^-10
o2 = e^21 / (e^0 + e^21) = 1

Feel free to try this example on your calculator.

**On the next page, you will be given a neural network representing scheduling behavior. Given input data about three tasks, you will have to use the neural network given to decide which task to schedule.**

**The input array will be of size 4. i1 will correspond to the first element, and so forth.**

**For example, for the input data of [11, 22, 33, 44, 55, 66, 77], i4 corresponds to 44.**

**Standard Neural Network Test**

**These page timer metrics will not be displayed to the recipient.**

#EditSection, TimingFirstClick#: *0 seconds*

#EditSection, TimingLastClick#: *0 seconds*

#EditSection, TimingPageSubmit#: *0 seconds*

#EditSection, TimingClickCount#: *0 clicks*

# Input data:

# [1, 2, 3, 1]

# The outputs of the neural network correspond to Task 1, Task 2, and Task 3.

**Note that the ReLU function is applied after the hidden layer. Simply, this function transforms any number below 0 to 0, and any other number stays the same. After computing o1, o2, o3, please apply the softmax function.**

$$W_1 = \begin{array}{c} \\ h1 \\ h2 \\ h3 \\ h4 \\ h5 \\ h6 \end{array} \begin{array}{cccc} i1 & i2 & i3 & i4 \\ \left[\begin{array}{cccc} 0 & -0.5 & 0 & -0.5 \\ 0 & -0.25 & 0 & 0 \\ 0 & 1.25 & -1 & 1.75 \\ -1 & 1 & 0.5 & -2.25 \\ 2 & -0.25 & -1.5 & 3.5 \\ 1 & 0.5 & -1.5 & -1.5 \end{array}\right]\end{array}$$

$$B_1 = \begin{array}{c} b1 \\ b2 \\ b3 \\ b4 \\ b5 \\ b6 \end{array} \left[\begin{array}{c} -0.25 \\ 0 \\ -0.5 \\ 0.5 \\ 0 \\ 0 \end{array}\right]$$

$$W_2 = \begin{array}{c} \\ j1 \\ j2 \\ j3 \\ j4 \\ j5 \\ j6 \end{array} \begin{array}{cccccc} h1 & h2 & h3 & h4 & h5 & h6 \\ \left[\begin{array}{cccccc} -0.5 & -0.25 & -1.75 & 3.5 & -4 & -0.25 \\ 0 & 0 & -1 & 0.75 & 3.25 & 2.75 \\ 0.25 & -0.25 & 2.25 & -0.75 & -0.25 & -0.25 \\ -0.25 & 0 & 1.75 & -0.25 & 2.75 & -2.5 \\ 0.25 & -0.5 & -1.75 & 3.5 & -4.5 & -0.25 \\ 0.25 & 0 & -2.75 & -2.25 & -0.75 & 1.5 \end{array}\right]\end{array}$$

$$B_2 = \begin{array}{c} b1 \\ b2 \\ b3 \\ b4 \\ b5 \\ b6 \end{array} \left[\begin{array}{c} 1.25 \\ 0 \\ 3 \\ -3.5 \\ 0.75 \\ 4.5 \end{array}\right]$$

$$W_3 = \begin{array}{c} \\ o1 \\ o2 \\ o3 \end{array} \begin{array}{cccccc} j1 & j2 & j3 & j4 & j5 & j6 \\ \left[\begin{array}{cccccc} 2.25 & -0.75 & -6 & 0 & 2.5 & -15.5 \\ -6.75 & -5.25 & 0.75 & 1 & -7 & 2 \\ -2 & 6 & -2.75 & -21.5 & 1.5 & -2 \end{array}\right]\end{array}$$

$$B_3 = \begin{array}{c} b1 \\ b2 \\ b3 \end{array} \left[\begin{array}{c} -3.5 \\ 0 \\ 0.5 \end{array}\right]$$

## Relu Function

$$f(x)= \begin{cases} 0 & \text{for } x < 0 \\ x & \text{for } x => 0 \end{cases}$$

Softmax Function

$$\text{Softmax}(x_i) = \frac{\exp(x_i)}{\sum_j \exp(x_j)}$$

Please fill in each question below.  Please round to 3 decimal places.

h1 (before ReLU)

h2 (before ReLU)

h3 (before ReLU)

h4 (before ReLU)

h5 (before ReLU)

h6 (before ReLU)

j1 (before ReLU)

j2 (before ReLU)

j3 (before ReLU)

j4 (before ReLU)

j5 (before ReLU)

j6 (before ReLU)

o1 (after softmax)

o2 (after softmax)

o3 (after softmax)

What is the output of this neural network given the input above? Please type the number associated with the output/task?

For example, if the output you receive is Task 5 has the highest score, you would type "5".

[                                                                  ]

**Standard Neural Network Survey**

Please fill out the questions below. The phrase "decision-making model" refers specifically to the graphic on the previous page. The phrase overall "decision-making process" refers to the entire process starting from being given input(s)  to answering the question(s).

| | Very Strongly Disagree | Strongly Disagree | Disagree | Neutral | Agree | Strongly Agree | Very Strongly Agree |
|---|---|---|---|---|---|---|---|
| The decision-making model is interpretable. | O | O | O | O | O | O | O |
| I understand the behavior represented within the decision-making model. | O | O | O | O | O | O | O |
| The decision-making model logic is easy to follow. | O | O | O | O | O | O | O |
| The decision-making model does not make sense. | O | O | O | O | O | O | O |
| The decision-making model is difficult to understand. | O | O | O | O | O | O | O |
| I could follow the rules of this decision-making model with ease. | O | O | O | O | O | O | O |
| I like the level of readability of this decision-making model. | O | O | O | O | O | O | O |

| | Very Strongly Disagree | Strongly Disagree | Disagree | Neutral | Agree | Strongly Agree | Very Strongly Agree |
|---|---|---|---|---|---|---|---|
| The overall decision-making process is easy to comprehend. | ◯ | ◯ | ◯ | ◯ | ◯ | ◯ | ◯ |
| I understand the overall process of choosing an output given input(s). | ◯ | ◯ | ◯ | ◯ | ◯ | ◯ | ◯ |
| This overall decision-making process logic is easy to follow. | ◯ | ◯ | ◯ | ◯ | ◯ | ◯ | ◯ |
| The overall decision-making process does not make sense. | ◯ | ◯ | ◯ | ◯ | ◯ | ◯ | ◯ |
| The overall decision-making process is difficult to understand. | ◯ | ◯ | ◯ | ◯ | ◯ | ◯ | ◯ |
| I could follow the rules of this decision-making tool with ease. | ◯ | ◯ | ◯ | ◯ | ◯ | ◯ | ◯ |

**Introduction Pointwise**

Hello,

Thank you for agreeing to do this survey!

For this survey, you will be asked to use 2 different types of decision-making frameworks and rate them based on their interpretability.

I will first provide you with instructions on how to use a certain type of decision-making framework. After this, I will ask you to click to the "next" button and use the decision-making framework to trace an input to an output. **I will ask that you do not take breaks on this page, as the speed at which you complete this exercise is timed**. At this point, you will be asked to answer some subjective questions about the decision-making framework used and the overall process of choosing a decision given the

method.

This process will be repeated two times for two different decision-making frameworks. This should take you approximately 35 minutes.

Please have a paper, writing utensil, and calculator at hand as you may need keep track of up to 10 numbers at once.  You may use a code editor as a calculator but please do not use it to assist with matrix multiplication. **As compensation, you will receive a $10 Amazon gift card.** Note that you must meet a minimum precalculated survey statistic to receive the compensation.

This survey has been approved by the Georgia Institute of Technology IRB. We recommend that you conduct this survey on your computer rather than on a mobile device.

**Pointwise Decision Tree Introduction**

A pointwise decision tree is a decision-making model in which each internal node represents a "test" on an attribute (e.g. whether a coin flip comes up heads or tails), each branch represents the outcome of the test, and each leaf node represents a score for the input. For this tree, you will be given a set of inputs, each corresponding to a specific output class. You must plug each input into the tree, and choose the input associated with the highest score.

In our trees, moving LEFT in the decision tree is associated a "test" of an attribute being true.

Let's do a quick example. Say you are given three inputs, representing some information about 3 people.
A: [Male, Age=2, SibSp=2, Pclass=3, Fare=4],
B: [Female, Age=2, SibSp=2, Pclass=2, Fare=4],
C:[Female, Age=2, SibSp=2, Pclass=3, Fare=4].

You put each input through the tree below and you will get scores of 0,1,0 for A,B,C respectively. We choose B as the output as it has the highest score.

**On the next page, you will be given a decision tree representing scheduling behavior. Given input data about the difficulty of three tasks and whether the scheduler prefers to handle difficult tasks or easy tasks, you will have to use the decision tree given to decide which task to schedule.**

**For example, for the input data of [11, 22, 33, 44, 55], {3} refers to 33.**

**Pointwise Decision Tree Test**

**These page timer metrics will not be displayed to the recipient.**
#EditSection, TimingFirstClick#: *0 seconds*
#EditSection, TimingLastClick#: *0 seconds*
#EditSection, TimingPageSubmit#: *0 seconds*
#EditSection, TimingClickCount#: *0 clicks*

**Input data:**

**Task 1: [3]**
**Task 2: [2]**
**Task 3: [1]**

**{2} fixed to 0.1**

**Clarification: The data for each task is feature {1}. {2} is fixed to 0.1.**

**The instructions from the previous page are found at the bottom of this page. Note that the previous page used Person 1, Person 2, and Person 3 and here we use Task 1, Task 2, and Task 3.**

Pointwise decision tree representing scheduling behavior.

Please write the scores of each task.

Task 1 Score

Task 2 Score

Task 3 Score

What is the output of this decision tree given the input above? Please type the number of the task. In the case of ties, break ties using numerical order.

For example, if the output you receive is task 5 has the highest score, you would type 5.

## Instructions for Pointwise DT

A pointwise decision tree is a decision-making model in which each internal node represents a "test" on an attribute (e.g. whether a coin flip comes up heads or tails), each branch represents the outcome of the test, and each leaf node represents a score for the input. For this tree, you will be given a set of inputs, each corresponding to a specific output class. You must plug each input into the tree, and choose the input associated with the highest score.

In our trees, moving LEFT in the decision tree is associated a "test" of an attribute being true.

Let's do a quick example. Say you are given three inputs,
A: [Male, Age=2, SibSp=2, Pclass=3, Fare=4],
B: [Female, Age=2, SibSp=2, Pclass=2, Fare=4],
C:[Female, Age=2, SibSp=2, Pclass=3, Fare=4].

You put each input through the tree below and you will get scores of 0,1,0 for A,B,C respectively. We choose B as the output as it has the highest score.

On the next page, you will be given a decision tree representing scheduling behavior. Given input data about the difficulty of three tasks and whether the scheduler prefers to handle difficult tasks or easy tasks, you will have to use the decision tree given to decide which task to schedule.

For example, for the input data of [11, 22, 33, 44, 55], {3} refers to 33.

## Pointwise Decision Tree Survey

Please fill out the questions below. The phrase "decision-making model" refers specifically to the graphic on the previous page. The phrase overall "decision-making process" refers to the entire process starting from being given input(s)  to answering the question(s).

| | Very Strongly Disagree | Strongly Disagree | Disagree | Neutral | Agree | Strongly Agree | Very Strongly Agree |
|---|---|---|---|---|---|---|---|
| The decision-making model is interpretable. | O | O | O | O | O | O | O |

| | Very Strongly Disagree | Strongly Disagree | Disagree | Neutral | Agree | Strongly Agree | Very Strongly Agree |
|---|---|---|---|---|---|---|---|
| I understand the behavior represented within the decision-making model. | ○ | ○ | ○ | ○ | ○ | ○ | ○ |
| The decision-making model logic is easy to follow. | ○ | ○ | ○ | ○ | ○ | ○ | ○ |
| The decision-making model does not make sense. | ○ | ○ | ○ | ○ | ○ | ○ | ○ |
| The decision-making model is difficult to understand. | ○ | ○ | ○ | ○ | ○ | ○ | ○ |
| I could follow the rules of this decision-making model with ease. | ○ | ○ | ○ | ○ | ○ | ○ | ○ |
| I like the level of readability of this decision-making model. | ○ | ○ | ○ | ○ | ○ | ○ | ○ |
| The overall decision-making process is easy to comprehend. | ○ | ○ | ○ | ○ | ○ | ○ | ○ |
| I understand the overall process of choosing an output given input(s). | ○ | ○ | ○ | ○ | ○ | ○ | ○ |
| This overall decision-making process logic is easy to follow. | ○ | ○ | ○ | ○ | ○ | ○ | ○ |
| The overall decision-making process does not make sense. | ○ | ○ | ○ | ○ | ○ | ○ | ○ |
| The overall decision-making process is difficult to understand. | ○ | ○ | ○ | ○ | ○ | ○ | ○ |
| I could follow the rules of this decision-making tool with ease. | ○ | ○ | ○ | ○ | ○ | ○ | ○ |

## Pointwise Neural Network Introduction

Here, we introduce pointwise neural networks.

**Please pay attention as you will not be allowed to come back and view this page.**

In the neural network below, the input into the neural network would be the input features [How much homework does the person do?, How many video games does the person play?, What is the work rate of the person?].

Say you are given three inputs,
Person 1: [Homework = 10, Video Games = 3, Utility = 1],
Person 2: [Homework = 3, Video Games = 4, Utility = 0],
Person 3: [Homework = 14, Video Games = 3, Utility = 14].

We can find the output of our network in matrix form using the equations:
$H = W\_1 * x + B\_1$
$H = ReLU(H)$
$O = W\_2 * H + B\_2$

$$W_1 = \begin{array}{c} \\ h1 \\ h2 \end{array} \begin{array}{ccc} i1 & i2 & i3 \\ \begin{bmatrix} 1 & 0 & 1 \\ 3 & 0 & 2 \end{bmatrix} \end{array}$$

$$B_1 = \begin{array}{c} b1 \\ b2 \end{array} \begin{bmatrix} 0 \\ 1 \end{bmatrix}$$

$$W_2 = \begin{array}{c} \\ o1 \end{array} \begin{array}{cc} h1 & h2 \\ \begin{bmatrix} 0 & 1 \end{bmatrix} \end{array}$$

$$B_2 = \begin{array}{c} b1 \end{array} \begin{bmatrix} 1 \end{bmatrix}$$

The scores of each person can be computed by inputting each into the neural network. To compute the output of Person 1 = [10, 3, 1],

To get to value of h1, we do (10 * 1 + 3 * 0 + 1 * 1) + 0 = 11
To get to value of h2, we do (10 * 3 + 3 * 0 + 1 * 2) + 1 = 33

Now we apply the ReLU activation function to h1 and h2 (shown below). Simply, this function transforms any number below 0 to 0, and any other number stays the same. Since h1 and h2 are positive, they remain unchanged.

To find o1, we do (11 * 0 + 33 * 1) + 1 = 34 to get the value score of o1.

We can do this for each input and put the outputs into the following vector form.

$$\begin{bmatrix} \text{Person 1} \\ \text{Person 2} \\ \text{Person 3} \end{bmatrix} = \begin{bmatrix} 34 \\ 11 \\ 72 \end{bmatrix}$$

Since Person C has the highest score, we choose person C.

**On the next page, you will be given a neural network representing scheduling behavior. Given input data about three tasks, you will have to use the neural network given to decide which task to schedule.**

**The input array will be of size 5. i1 will correspond to the first element, and so forth.**

**For example, for the input data of [11, 22, 33, 44, 55, 66, 77], i4 corresponds to 44.**

**Pointwise Neural Network Test**

**These page timer metrics will not be displayed to the recipient.**
#EditSection, TimingFirstClick#: *0 seconds*
#EditSection, TimingLastClick#: *0 seconds*
#EditSection, TimingPageSubmit#: *0 seconds*
#EditSection, TimingClickCount#: *0 clicks*

 **Input data:**

**Task 1: [1]**
**Task 2: [2]**
**Task 3: [3]**

# The instructions found on the previous page are attached at the end of this page for reference.

Pointwise neural network representing scheduling behavior. A ReLU is applied after layer 1.

## i2 is fixed to 1.

$$W_1 = \begin{array}{c} \\ h1 \\ h2 \\ h3 \\ h4 \\ h5 \end{array} \begin{array}{cc} i1 & i2 \\ \left[\begin{array}{cc} 0.25 & -0.5 \\ -0.25 & -0.75 \\ -0.25 & -0.5 \\ -2.25 & -0.25 \\ -0.75 & 3.25 \end{array}\right] \end{array}$$

$$B_1 = \begin{array}{c} b1 \\ b2 \\ b3 \\ b4 \\ b5 \end{array} \left[\begin{array}{c} -0.5 \\ -0.25 \\ 0 \\ 2 \\ -0.75 \end{array}\right]$$

$$W_2 = \begin{array}{c} \\ o1 \end{array} \begin{array}{ccccc} j1 & j2 & j3 & j4 & j5 \\ \left[\begin{array}{ccccc} -0.5 & -0.25 & -0.25 & 2 & -5 \end{array}\right] \end{array}$$

$$B_2 = \begin{array}{c} b1 \end{array} \left[\begin{array}{c} -0.25 \end{array}\right]$$

Please write the final scores below.

Task 1 Score ___________

Task 2 Score ___________

Task 3 Score ___________

What is the output of this neural network given the input above? Please type the number of the task. In the case of ties, break ties using numerical order.

For example, if the output you receive is task 5 has the highest score, you would type 5.

___________

Here, we introduce pointwise neural networks.

**Please pay attention as you will not be allowed to come back and view this page.**

In the neural network below, the input into the neural network would be the input features [How much homework does the person do?, How many video games does the person play?, What is the work rate of the person?].

Say you are given three inputs,
Person 1: [Homework = 10, Video Games = 3, Utility = 1],
Person 2: [Homework = 3, Video Games = 4, Utility = 0],
Person 3: [Homework = 14, Video Games = 3, Utility = 14].

We can find the output of our network in matrix form using the equations:
$H = W\_1 * x + B\_1$
$H = ReLU(H)$
$O = W\_2 * H + B\_2$

$$W_1 = \begin{array}{cc} & \begin{array}{ccc} i1 & i2 & i3 \end{array} \\ \begin{array}{c} h1 \\ h2 \end{array} & \begin{bmatrix} 1 & 0 & 1 \\ 3 & 0 & 2 \end{bmatrix} \end{array}$$

$$B_1 = \begin{array}{c} b1 \\ b2 \end{array} \begin{bmatrix} 0 \\ 1 \end{bmatrix}$$

$$W_2 = \begin{array}{c} \\ o1 \end{array} \begin{array}{cc} h1 & h2 \\ \begin{bmatrix} 0 & 1 \end{bmatrix} \end{array}$$

$$B_2 = \begin{array}{c} b1 \end{array} \begin{bmatrix} 1 \end{bmatrix}$$

The scores of each person can be computed by inputting each into the neural network. To compute the output of Person 1 = [10, 3, 1],

To get to value of h1, we do (10 * 1 + 3 * 0 + 1 * 1) + 0 = 11
To get to value of h2, we do (10 * 3 + 3 * 0 + 1 * 2) + 1 = 33

Now we apply the ReLU activation function to h1 and h2 (shown below). Simply, this function transforms any number below 0 to 0, and any other number stays the same. Since h1 and h2 are positive, they remain unchanged.

To find o1, we do (11 * 0 + 33 * 1) + 1 = 34 to get the value score of o1.

We can do this for each input and put the outputs into the following vector form.

$$\begin{bmatrix} \text{Person 1} \\ \text{Person 2} \\ \text{Person 3} \end{bmatrix} = \begin{bmatrix} 34 \\ 11 \\ 72 \end{bmatrix}$$

Since Person C has the highest score, we choose person C.

On the next page, you will be given a neural network representing scheduling behavior. Given input data about three tasks, you will have to use the neural network given to decide which task to schedule.

The input array will be of size 5. i1 will correspond to the first element, and so forth.

For example, for the input data of [11, 22, 33, 44, 55, 66, 77], i4 corresponds to 44.

**Pointwise Neural Network Survey**

Please fill out the questions below. The phrase "decision-making model" refers specifically to the graphic on the previous page. The phrase overall "decision-making process" refers to the entire process starting from being given input(s) to answering the question(s).

| | Very Strongly Disagree | Strongly Disagree | Disagree | Neutral | Agree | Strongly Agree | Very Strongly Agree |
|---|---|---|---|---|---|---|---|
| The decision-making model is interpretable. | O | O | O | O | O | O | O |

| | Very Strongly Disagree | Strongly Disagree | Disagree | Neutral | Agree | Strongly Agree | Very Strongly Agree |
|---|---|---|---|---|---|---|---|
| I understand the behavior represented within the decision-making model. | O | O | O | O | O | O | O |
| The decision-making model logic is easy to follow. | O | O | O | O | O | O | O |
| The decision-making model does not make sense. | O | O | O | O | O | O | O |
| The decision-making model is difficult to understand. | O | O | O | O | O | O | O |
| I could follow the rules of this decision-making model with ease. | O | O | O | O | O | O | O |
| I like the level of readability of this decision-making model. | O | O | O | O | O | O | O |
| The overall decision-making process is easy to comprehend. | O | O | O | O | O | O | O |
| I understand the overall process of choosing an output given input(s). | O | O | O | O | O | O | O |
| This overall decision-making process logic is easy to follow. | O | O | O | O | O | O | O |
| The overall decision-making process does not make sense. | O | O | O | O | O | O | O |
| The overall decision-making process is difficult to understand. | O | O | O | O | O | O | O |
| I could follow the rules of this decision-making tool with ease. | O | O | O | O | O | O | O |

## Introduction Pairwise

Hello,

Thank you for agreeing to do this survey!

For this survey, you will be asked to use 2 different types of decision-making frameworks and rate them based on their interpretability.

I will first provide you with instructions on how to use a certain type of decision-making framework. After this, I will ask you to click to the "next" button and use the decision-making framework to trace an input to an output. **I will ask that you do not take breaks on this page, as the speed at which you complete this exercise is timed**. At this point, you will be asked to answer some subjective questions about the decision-making framework used and the overall process of choosing a decision given the method.

This process will be repeated two times for two different decision-making frameworks. This should take you approximately 1 hour.

Please have a paper, writing utensil, and calculator at hand as you may need keep track of up to 10 numbers at once. You may use a code editor as a calculator but please do not use it to assist with matrix multiplication. **As compensation, you will receive a $15 Amazon gift card.** Note that you must meet a minimum precalculated survey statistic to receive the compensation.

This survey has been approved by the Georgia Institute of Technology IRB. We recommend that you conduct this survey on your computer rather than on a mobile device.

**Pairwise Decision Tree Introduction**

A pairwise decision tree is is decision-making model in which each internal node represents a "test" on an attribute (e.g. how much more homework person A completed than person B), each branch represents the outcome of the test, and each leaf node represents a **score for the difference of inputs** (e.g., how much higher person A's test grade is compared to person B). The paths from root to leaf represent classification rules.

In our trees, moving LEFT in the decision tree is associated a "test" of an attribute being true.

Let's do a quick example.

**Please pay attention as you will not be allowed to come back and view this page.**

Pairwise Decision Tree representing the decision score of "How much do you prefer person A compared to person B?".

In the decision tree above, the input into the decision tree would be a **set** of [How much homework does the person do?, How many video games does the person

play?, What is the utility of the person?]. To generate the decision tree input, we would subtract the traits of the first person from the second. Then, using this as the input to the decision tree, we can find a difference score for how much the first person is preferred compared to the second.

For example,

Say you are given three inputs,
Person 1: [Homework = 10, Video Games = 3, Utility = 1],
Person 2: [Homework = 3, Video Games = 4, Utility = 0],
Person 3: [Homework = 14, Video Games = 3, Utility = 14].

There are 6 combinations of subtracted vectors: (Person 1 - Person 3), (Person 2 - Person 3), (Person 3 - Person 1), (Person 3 - Person 2), (Person 1 - Person 2), and (Person 2 - Person 1).

Person 1 - Person 2 = [10, 3, 1] - [3, 4, 0] = [7, -1, 1]. Putting this through the decision tree generates a score of 1.

These can be put into a matrix form, where the element being subtracted from is the row, and the subtractor is the column.

$$\begin{bmatrix} 0 & \text{Person 1 - Person 2} & \text{Person 1 - Person 3} \\ \text{Person 2 - Person 1} & 0 & \text{Person 2 - Person 3} \\ \text{Person 3 - Person 1} & \text{Person 3 - Person2} & 0 \end{bmatrix}$$

Plugging in each subtracted pair into the decision tree above will produce this matrix.

$$\begin{bmatrix} 0 & 1 & 0.4 \\ 0.4 & 0 & 0.4 \\ 0.75 & 1 & 0 \end{bmatrix}$$

Then, we can sum across columns to get the score for each person , respectively.

$$\begin{bmatrix} \text{Person 1} \\ \text{Person 2} \\ \text{Person 3} \end{bmatrix} = \begin{bmatrix} 1.4 \\ 0.8 \\ 1.75 \end{bmatrix}$$

Since Person C has the highest score, we choose person C.

**On the next page, you will be given a decision tree representing scheduling behavior. Given input data about the difficulty of three tasks and whether the scheduler prefers to handle difficult tasks or easy tasks, you will have to use the decision tree given to decide which task to schedule.**

**The input array will be of size 3, and the decision tree will reference the corresponding element using {#} notation.**

**For example, for the input data of [11, 22, 33, 44, 55], {3} refers to 33.**

**Pairwise Decision Tree Test**

**These page timer metrics will not be displayed to the recipient.**
#EditSection, TimingFirstClick#: *0 seconds*
#EditSection, TimingLastClick#: *0 seconds*
#EditSection, TimingPageSubmit#: *0 seconds*
#EditSection, TimingClickCount#: *0 clicks*

# Input data:

**Task 1: [3]**
**Task 2: [2]**
**Task 3: [1]**

**{2} is fixed to 0.4,**

**Clarification: To compute the input feature {1} for (Task 1 - Task 2), we would subtract Task 1 - Task 2. {2} is fixed to 0.4.**

**The instructions from the previous page are found at the bottom of this page. Note that the previous page used Person 1, Person 2, and Person 3 and here we use Task 1, Task 2, and Task 3.**

Pairwise decision tree representing scheduling behavior.

Please write down the difference vectors (in the format [#]). For example, if task 5 had data of [4], and task 6 had data of [3], task 5 - task 6 is equal to [1].

Task 1 - Task 2

Task 1 - Task 3

Task 2 - Task 1

Task 2 - Task 3

Task 3 - Task 1

Task 3 - Task 2

Please fill in the fields below.

|  | Task 1 | Task 2 | Task 3 |
| --- | --- | --- | --- |
| Task 1 |  |  |  |
| Task 2 |  |  |  |
| Task 3 |  |  |  |

Please write the scores below.

Task 1 Score

Task 2 Score

Task 3 Score

What is the output of this decision tree given the input above? Please type the number of the task. In the case of ties, break ties using numerical order.

For example, if the output you receive is Task 5 has the highest score, you would type 5.

Instructions

A pairwise decision tree is is decision-making model in which each internal node represents a "test" on an attribute (e.g. how much more homework person A completed than person B), each branch represents the outcome of the test, and each leaf node represents a **score for the difference of inputs** (e.g., how much higher person A's test grade is compared to person B). The paths from root to leaf represent classification rules.

In our trees, moving LEFT in the decision tree is associated a "test" of an attribute being true.

Let's do a quick example.

**Please pay attention as you will not be allowed to come back and view this page.**

---

Pairwise Decision Tree representing the decision score of "How much do you prefer person A compared to person B?".

In the decision tree above, the input into the decision tree would be a **set** of [How much homework does the person do?, How many video games does the person play?, What is the utility of the person?]. To generate the decision tree input, we would subtract the traits of the first person from the second. Then, using this as the input to the decision tree, we can find a difference score for how much the first person is preferred compared to the second.

For example,

Say you are given three inputs,
Person 1: [Homework = 10, Video Games = 3, Utility = 1],
Person 2: [Homework = 3, Video Games = 4, Utility = 0],
Person 3: [Homework = 14, Video Games = 3, Utility = 14].

There are 6 combinations of subtracted vectors: (Person 1 - Person 3), (Person 2 - Person 3), (Person 3 - Person 1), (Person 3 - Person 2), (Person 1 - Person 2), and (Person 2 - Person 1).

Person 1 - Person 2 = [10, 3, 1] - [3, 4, 0] = [7, -1, 1]. Putting this through the decision tree generates a score of 1.

These can be put into a matrix form, where the element being subtracted from is the row, and the subtractor is the column.

$$
\begin{bmatrix}
0 & \text{Person 1 - Person 2} & \text{Person 1 - Person 3} \\
\text{Person 2 - Person 1} & 0 & \text{Person 2 - Person 3} \\
\text{Person 3 - Person 1} & \text{Person 3 - Person2} & 0
\end{bmatrix}
$$

Plugging in each subtracted pair into the decision tree above will produce this matrix.

$$
\begin{bmatrix}
0 & 1 & 0.4 \\
0.4 & 0 & 0.4 \\
0.75 & 1 & 0
\end{bmatrix}
$$

Then, we can sum across columns to get the score for each person , respectively.

$$
\begin{bmatrix}
\text{Person 1} \\
\text{Person 2} \\
\text{Person 3}
\end{bmatrix}
=
\begin{bmatrix}
1.4 \\
0.8 \\
1.75
\end{bmatrix}
$$

Since Person C has the highest score, we choose person C.

---

On the next page, you will be given a decision tree representing scheduling behavior. Given input data about the difficulty of three tasks and whether the scheduler prefers to handle difficult tasks or easy tasks, you will have to use the decision tree given to decide which task to schedule.

The input array will be of size 3, and the decision tree will reference the corresponding element using {#} notation.

For example, for the input data of [11, 22, 33, 44, 55], {3} refers to 33.

## Pairwise Decision Tree Survey

Please fill out the questions below. The phrase "decision-making model" refers specifically to the graphic on the previous page. The phrase overall "decision-making process" refers to the entire process starting from being given input(s) to answering the question(s).

| | Very Strongly Disagree | Strongly Disagree | Disagree | Neutral | Agree | Strongly Agree | Very Strongly Agree |
|---|---|---|---|---|---|---|---|
| The decision-making model is interpretable. | O | O | O | O | O | O | O |
| I understand the behavior represented within the decision-making model. | O | O | O | O | O | O | O |
| The decision-making model logic is easy to follow. | O | O | O | O | O | O | O |
| The decision-making model does not make sense. | O | O | O | O | O | O | O |
| The decision-making model is difficult to understand. | O | O | O | O | O | O | O |

| | Very Strongly Disagree | Strongly Disagree | Disagree | Neutral | Agree | Strongly Agree | Very Strongly Agree |
|---|---|---|---|---|---|---|---|
| I could follow the rules of this decision-making model with ease. | ○ | ○ | ○ | ○ | ○ | ○ | ○ |
| I like the level of readability of this decision-making model. | ○ | ○ | ○ | ○ | ○ | ○ | ○ |
| The overall decision-making process is easy to comprehend. | ○ | ○ | ○ | ○ | ○ | ○ | ○ |
| I understand the overall process of choosing an output given input(s). | ○ | ○ | ○ | ○ | ○ | ○ | ○ |
| This overall decision-making process logic is easy to follow. | ○ | ○ | ○ | ○ | ○ | ○ | ○ |
| The overall decision-making process does not make sense. | ○ | ○ | ○ | ○ | ○ | ○ | ○ |
| The overall decision-making process is difficult to understand. | ○ | ○ | ○ | ○ | ○ | ○ | ○ |
| I could follow the rules of this decision-making tool with ease. | ○ | ○ | ○ | ○ | ○ | ○ | ○ |

**Pairwise Neural Network Introduction**

Here, we introduce pairwise neural networks.

**Please pay attention as you will not be allowed to come back and view this page.**

In the neural network below, the input into the neural network would be a **set** of the input features [How much homework does the person do?, How many video games does the person play?, What is the work rate of the person?]. To generate the neural

network input, we would subtract the traits of the first person from the second. Then, using this as the input to the neural network, we can find a difference score for how much the first person is preferred to the second.

For example,

Say you are given three inputs,
Person 1: [Homework = 10, Video Games = 3, Utility = 1],
Person 2: [Homework = 3, Video Games = 4, Utility = 0],
Person 3: [Homework = 14, Video Games = 3, Utility = 14].

There are 6 combinations of subtracted vectors that will serve as input into the neural network: (Person 1 - Person 3), (Person 2 - Person 3), (Person 3 - Person 1), (Person 3 - Person 2), (Person 1 - Person 2), and (Person 2 - Person 1).

Person 1 - Person 2 = [10, 3, 1] - [3, 4, 0] = [7, -1, 1].
Person 1 - Person 3 = [10, 3, 1] - [14, 3, 14] = [-4, 0, -13]
and so on.

We can find the output of our network in matrix form using the equations, where the input, x, is the subtracted vectors described above:
$H = W\_1 * x + B\_1$
$H = ReLU(H)$
$O = W\_2 * H + B\_2$

$$W_1 = \begin{array}{c} \\ h1 \\ h2 \end{array} \begin{array}{ccc} i1 & i2 & i3 \\ \left[\begin{array}{ccc} 1 & 0 & 1 \\ 3 & 0 & 2 \end{array}\right] \end{array}$$

$$B_1 = \begin{array}{c} b1 \\ b2 \end{array} \left[\begin{array}{c} 0 \\ 1 \end{array}\right]$$

$$W_2 = \begin{array}{c} \\ o1 \end{array} \begin{array}{cc} h1 & h2 \\ \left[\begin{array}{cc} 0 & 1 \end{array}\right] \end{array}$$

$$B_2 = \begin{array}{c} b1 \end{array} \left[\begin{array}{c} 1 \end{array}\right]$$

Computing the output of Person 1 - Person 2 = [10, 3, 1] - [3, 4, 0] = [7, -1, 1]:

To get to value of h1, we do (7 * 1 + -1 * 0 + 1 * 1) + 0 = 8
To get to value of h2, we do (7 * 3 + -1 * 0 + 1 * 2) + 1 = 24

Now we apply the ReLU activation function to h1 and h2 (shown below). Simply, this function transforms any number below 0 to 0, and any other number stays the same. Since h1 and h2 are positive, they remain unchanged.

To find o1, we do (8 * 0 + 24 * 1) + 1 = 25 to get the value score of o1.

We can do this for each input and put the outputs into the following matrix form.

$$\begin{bmatrix} 0 & \text{Person 1 - Person 2} & \text{Person 1 - Person 3} \\ \text{Person 2 - Person 1} & 0 & \text{Person 2 - Person 3} \\ \text{Person 3 - Person 1} & \text{Person 3 - Person2} & 0 \end{bmatrix}$$

The final output matrix is then

$$\begin{bmatrix} 0 & 25 & 0 \\ 0 & 0 & 0 \\ 40 & 63 & 0 \end{bmatrix}$$

Then, we can sum across columns to get the score for each person , respectively.

$$\begin{bmatrix} \text{Person 1} \\ \text{Person 2} \\ \text{Person 3} \end{bmatrix} = \begin{bmatrix} 25 \\ 0 \\ 103 \end{bmatrix}$$

Since Person C has the highest score, we choose person C.

**On the next page, you will be given a neural network representing scheduling behavior. Given input data about three tasks, you will have to use the neural network given to decide which task to schedule.**

**The input array will be of size 2. i1 will correspond to the first element, and so forth.**

**For example, for the input data of [11, 22, 33, 44, 55, 66, 77], i4 corresponds to 44.**

**Pairwise Neural Network Test**

**These page timer metrics will not be displayed to the recipient.**
#EditSection, TimingFirstClick#: *0 seconds*
#EditSection, TimingLastClick#: *0 seconds*
#EditSection, TimingPageSubmit#: *0 seconds*
#EditSection, TimingClickCount#: *0 clicks*

**Input data:**

**Task 1: [1]**
**Task 2: [2]**
**Task 3: [3]**

**Clarification: To compute the input feature for (Task 1 - Task 2), we would subtract Task 1 - Task 2 to get i1. i2 is fixed to the values mentioned below.**

**The instructions found on the previous page are attached at the end of this page for reference.**

Pairwise neural network representing scheduling behavior. A ReLU is applied after the first layer.

## i2 is fixed to 1.

$$
\begin{array}{c}
\\
h1 \\
h2 \\
h3 \\
h4 \\
h5
\end{array}
\begin{array}{cc}
i1 & i2 \\
\left[\begin{array}{cc}
0.25 & -0.5 \\
-0.25 & -1.5 \\
0 & -0.5 \\
-0.5 & -2.25 \\
0.5 & 2.50
\end{array}\right]
\end{array}
$$

$$
\begin{array}{c}
b1 \\
b2 \\
b3 \\
b4 \\
b5
\end{array}
\left[\begin{array}{c}
-0.5 \\
0.5 \\
0 \\
0.75 \\
-1.5
\end{array}\right]
$$

$$
\begin{array}{cccccc}
 & j1 & j2 & j3 & j4 & j5 \\
o1 & [0.25 & 1.75 & -0.25 & 2.75 & 2]
\end{array}
$$

$$
b1 \quad [-1.5]
$$

**Please write down the difference vectors (in the format [#]). For example, if task 5 had data of [4], and task 6 had data of [3], task 5 - task 6 is equal to [1].**

| | |
|---|---|
| Task 1 - Task 2 | |
| Task 1 - Task 3 | |
| Task 2 - Task 1 | |
| Task 2 - Task 3 | |
| Task 3 - Task 1 | |
| Task 3 - Task 2 | |

Please fill out the fields below.

| | Task 1 | Task 2 | Task 3 |
|---|---|---|---|
| Task 1 | | | |
| Task 2 | | | |
| Task 3 | | | |

Please write the final scores below.

| | |
|---|---|
| Task 1 Score | |
| Task 2 Score | |
| Task 3 Score | |

What is the output of this neural network given the input above? Please type the number of the task. In the case of ties, break ties using numerical order.

For example, if the output you receive is task 5 has the highest score, you would type 5.

Here are the instructions for reference.

Here, we introduce pairwise neural networks.

**Please pay attention as you will not be allowed to come back and view this page.**

In the neural network below, the input into the neural network would be a **set** of the input features [How much homework does the person do?, How many video games does the person play?, What is the work rate of the person?]. To generate the decision tree input, we would subtract the traits of the first person from the second. Then, using this as the input to the decision tree, we can find a difference score for how much the first person is preferred to the second.

For example,

Say you are given three inputs,
Person 1: [Homework = 10, Video Games = 3, Utility = 1],
Person 2: [Homework = 3, Video Games = 4, Utility = 0],
Person 3: [Homework = 14, Video Games = 3, Utility = 14].

There are 6 combinations of subtracted vectors that will serve as input into the neural network: (Person 1 - Person 3), (Person 2 - Person 3), (Person 3 - Person 1), (Person 3 - Person 2), (Person 1 - Person 2), and (Person 2 - Person 1).

Person 1 - Person 2 = [10, 3, 1] - [3, 4, 0] = [7, -1, 1].
Person 1 - Person 3 = [10, 3, 1] - [14, 3, 14] = [-4, 0, -13]
and so on.

We can find the output of our network in matrix form using the equations:

$H = W\_1 * x + B\_1$

$H = ReLU(H)$

$O = W\_2 * H + B\_2$

$$W_1 = \begin{array}{c} \\ h1 \\ h2 \end{array} \begin{array}{ccc} i1 & i2 & i3 \\ \begin{bmatrix} 1 & 0 & 1 \\ 3 & 0 & 2 \end{bmatrix} \end{array}$$

$$B_1 = \begin{array}{c} b1 \\ b2 \end{array} \begin{bmatrix} 0 \\ 1 \end{bmatrix}$$

$$W_2 = \begin{array}{c} \\ o1 \end{array} \begin{array}{cc} h1 & h2 \\ \begin{bmatrix} 0 & 1 \end{bmatrix} \end{array}$$

$$B_2 = \begin{array}{c} b1 \end{array} \begin{bmatrix} 1 \end{bmatrix}$$

Computing the output of Person 1 - Person 2 = [10, 3, 1] - [3, 4, 0] = [7, -1, 1].

To get to value of h1, we do (7 * 1 + -1 * 0 + 1 * 1) + 0 = 8
To get to value of h2, we do (7 * 3 + -1 * 0 + 1 * 2) + 1 = 24

Now we apply the ReLU activation function to h1 and h2 (shown below). Simply, this function transforms any number below 0 to 0, and any other number stays the same. Since h1 and h2 are positive, they remain unchanged.

To find o1, we do (8 * 0 + 24 * 1) + 1 = 25 to get the value score of o1.

We can do this for each input and put the outputs into the following matrix form.

$$\begin{bmatrix} 0 & \text{Person 1 - Person 2} & \text{Person 1 - Person 3} \\ \text{Person 2 - Person 1} & 0 & \text{Person 2 - Person 3} \\ \text{Person 3 - Person 1} & \text{Person 3 - Person2} & 0 \end{bmatrix}$$

The final output matrix is then

$$\begin{bmatrix} 0 & 25 & 0 \\ 0 & 0 & 0 \\ 40 & 63 & 0 \end{bmatrix}$$

Then, we can sum across columns to get the score for each person , respectively.

$$\begin{bmatrix} \text{Person 1} \\ \text{Person 2} \\ \text{Person 3} \end{bmatrix} = \begin{bmatrix} 25 \\ 0 \\ 103 \end{bmatrix}$$

Since Person C has the highest score, we choose person C.

**On the next page, you will be given a neural network representing scheduling behavior. Given input data about three tasks, you will have to use the neural network given to decide which task to schedule.**

**The input array will be of size 5. i1 will correspond to the first element, and so forth.**

**For example, for the input data of [11, 22, 33, 44, 55, 66, 77], i4 corresponds to 44.**

## Pairwise Neural Network Survey

Please fill out the questions below. The phrase "decision-making model" refers specifically to the graphic on the previous page. The phrase overall "decision-making process" refers to the entire process starting from being given input(s)  to answering the question(s).

| | Very Strongly Disagree | Strongly Disagree | Disagree | Neutral | Agree | Strongly Agree | Very Strongly Agree |
|---|---|---|---|---|---|---|---|
| The decision-making model is interpretable. | O | O | O | O | O | O | O |
| I understand the behavior represented within the decision-making model. | O | O | O | O | O | O | O |
| The decision-making model logic is easy to follow. | O | O | O | O | O | O | O |

| | Very Strongly Disagree | Strongly Disagree | Disagree | Neutral | Agree | Strongly Agree | Very Strongly Agree |
|---|---|---|---|---|---|---|---|
| The decision-making model does not make sense. | O | O | O | O | O | O | O |
| The decision-making model is difficult to understand. | O | O | O | O | O | O | O |
| I could follow the rules of this decision-making model with ease. | O | O | O | O | O | O | O |
| I like the level of readability of this decision-making model. | O | O | O | O | O | O | O |
| The overall decision-making process is easy to comprehend. | O | O | O | O | O | O | O |
| I understand the overall process of choosing an output given input(s). | O | O | O | O | O | O | O |
| This overall decision-making process logic is easy to follow. | O | O | O | O | O | O | O |
| The overall decision-making process does not make sense. | O | O | O | O | O | O | O |
| The overall decision-making process is difficult to understand. | O | O | O | O | O | O | O |
| I could follow the rules of this decision-making tool with ease. | O | O | O | O | O | O | O |

## Conclusion

Thank you for agreeing to do this survey. Your response will help us generate interpretable decision-making models.

Please leave your email below and we will send you an Amazon gift card. If you have any further questions, please email Rohan Paleja at rpaleja3@gatech.edu.

Email Address

Comments on our survey?

Powered by Qualtrics



[Supplementary Material 2]

# Supplementary for Interpretable and Personalized Apprenticeship Scheduling: Learning Interpretable Scheduling Policies from Heterogeneous User Demonstrations

## 1 Additional Experiment Domain Details

**Synthetic Scheduling Environment**    The synthetic scheduling environment represents one of the hardest scheduling problems. In this environment, two agents must complete a set of 20 tasks which have upper- and lower-bound temporal constraints (i.e., deadline and wait constraints), proximity constraints (i.e., no two agents can be in the same place at the same time), and travel-time constraints. For the purposes of apprenticeship learning, an action is defined as the assignment of an agent to complete a task presently. The decision-maker must decide the optimal sequence of actions according to the decision-maker's own criteria. For this environment, we construct a set of heterogeneous, mock decision-makers that schedule according to Equation 1.

$$\tau_i^* = \underset{\tau_j \subset \boldsymbol{\tau_S}}{\arg\max}(\rho_1 H_{EDF}(\tau_j) + \rho_2 H_{dist}(\tau_j) + H_{ID}(\tau_j, \rho_3)) \tag{1}$$

In this equation, our decision-maker selects a task $\tau_i^*$ from the set of tasks $\boldsymbol{\tau_S}$. The task-prioritization scheme is based on three criteria: $H_{EDF}$ prioritizes tasks according to deadline (i.e., "earliest-deadline first"), $H_{dist}$ prioritizes the closest task, and $H_{ID}$ prioritizes tasks according to a user-specified highest/lowest index or value based on $\rho_3$ (i.e., $\rho_3(j) + (1 - \rho_3)(-j)$). The heterogeneity in decision-making comes from the latent weighting vector $\vec{\rho}$. Specifically, $\rho_1 \in \mathbb{R}$ and $\rho_2 \in \mathbb{R}$ weight the importance of $H_{EDF}$ and $H_{dist}$, respectively. $\rho_3 \in \{0, 1\}$ is a mode selector in which the highest/lowest task index is prioritized. By drawing $\vec{\rho}$ from a multivariate random distribution, we can create an infinite number of unique demonstrator types. This adapted environment differs from the synthetic, low-dimensional environment in that there are a rich set of temporal, spatial, and agent-based constraints modeling the job-shop scheduling problem; furthermore, the parameters of the demonstrator's decision-making process is hidden and comprised of one discrete factor and two continuous factors. In this domain, counterfactuals are generated by consider specific task information such as availability, distance from agent, prerequisites satisfied.

**Real-world Data: Taxi Domain**    Our environment has three locations: the village, the airport, and the city. The taxi driver has the objective of picking up a passenger from the city or village. There is always a passenger at the city, but the taxi driver may encounter up to 60 minutes of traffic going into the city. There may be a wait time of up to 60 minutes to pick up a passenger at the village; however, there is no traffic on the way to the village, and the wait time is unknown to the taxi driver unless she is at the village. A dataset of 70 human-collected tree policies to solve this task (given with leaf node actions such as "Drive to the City", "Drive to the Airport", and "Wait for Passenger", and decision node criterion depending on the amount of wait time, traffic time, and current location) are used to generate heterogeneous trajectories. We originally collect 98 tree-based policies through an IRB-approved study. However, 28 of these do not produce successful trajectories. The tree-based policies

Table 1: Apprenticeship Performance in Imitating Robot Kinesthetic Ping Pong Demonstrations.

| Environment | Our Method | Sammut et al. | Nikolaidis et al. | Tamar et al. | Hsiao et. al. | InfoGAIL Li et. al. | Gombolay et. al. |
|---|---|---|---|---|---|---|---|
| Ping-Pong | **59.60%** | 18.14% | 31.20% | 26.17% | 17.96% | 36.70% | 28.60% |

can be found in this GitHub repository https://github.com/Personalized-Neural-Trees/Interpretable-and-Personalized-Apprenticeship-Scheduling-Learning-Interpretable-Scheduling-Policies.

**Kinesthetic Robot Table Tennis**    We collected a real-world data set consisting of 10 human demonstrators kinesthetically presenting four different table tennis strikes on a Rethink Robotics Sawyer. The table tennis strike variants consisted of push, topspin, slice, and lob and were conducted using a forehand motion, giving four different categories of motion. While our approach is primarily for discrete classification problems, such as decision making, it can naturally be extended to complex continuous domains, such as low-level robot joint control.

To collect data, we first show each demonstrator a sample video of the table tennis strike and allow them to practice until she feels confident that she can return the ping pong ball over the net. Then, we reset the robotic arm to a preset initial position and allow the demonstrator to strike a ping pong ball launched from an automatic ball launcher. Throughout the demonstration, we record the position of the end-effector.

**Survey Scheduling Environment**    This domain describes a **ND[ST-SR-TA]** scheduling domain defined by Korsah [5]. In this domain, synthetic schedulers are given utilities of three tasks where utility $U \in \{1, 2, 3\}$ and must choose the highest or lowest task index based on a pre-specified latent decision-making criteria. We generate a set of 100 schedules (each of length 20) from heterogeneous demonstrators.

# 2   LfD Performance in Kinesthetic Robot Table Tennis

Here, we show that Personalized Neural Trees can easily be extended to a variety of domains, increasing the data-efficiency, accuracy, and utility of learning-from-demonstration with multiple human demonstrators. We demonstrate this by using a PNT to learn kinesthetic robot table tennis demonstrations in Table 1. We received 40 demonstrations across 10 demonstrators, representing four different table tennis strikes. To clean the trajectory of the end effector prior to learning, we transformed our trajectories into a transformed three-dimensional space using Principal Components Analysis. Our data was then labeled by selecting the principal component in which the end effector moved most at each timestep ($|\mathcal{A}| = 6$). As seen in row 4 of Table 1, our approach outperforms all other benchmarks.

# 3   Sensitivity Analysis of PNTs

To analyze the sensitivity of our framework, we use our synthetic scheduling environment and perturb the amount of data available to the PNT and the amount of noise (correctness) within the data. To provide a thorough analysis, we validate our approach using $k$-fold cross-validation. This entails both choosing a different subset of data to learn from and perturbing different truth-values of state-actions pairs each fold.

As shown in Figure 1, our PNT is reasonably robust to noise for 2, 5, and 15 schedules as there is not a steep drop in accuracy. We do not see the typical trend where the effect of noise deteriorates as the amount of data increases. We posit the cause of this deviation as follows: As the number of demonstrators increases, the embedding space $\Omega$ of the PNT tends to represent a richer distribution. While the heterogeneity among the demonstrators may remain constant (represent the same number of modes), cases in which the PNT is unable to tease out the demonstrator mode from a single schedule are more likely (due to the increase in the number of schedules), leading to an embedding distribution with higher variance. Without noise, the PNT is able to make sense of the embedding space and learn with high performance; as the amount of noise increases, it is likely more difficult to

Figure 1: Sensitivity analysis in the synthethic scheduling environment.

represent demonstrators compactly within the embedding space. We posit that this increased variance
within the embedding space caused by the combined effect of an increased number of demonstrators
and noise leads to a reduction in performance when noise is held constant and the amount of data
increases.

As expected, as the number of schedules increase, the PNTs have higher accuracy. However, from
15 to 150 schedules (a 10x magnitude increase in data), for the case of 100% correct data, there is
only a $\sim 2\%$ increase in accuracy. This result provides support to the claim of data-efficiency in our
apprenticeship scheduling framework.

# 4 Evidence Lower Bound

Here, we present the full derivation of the evidence lower bound (ELBO) that is used maximize the
mutual information between $\omega$ and trajectories $\tau$.

$$
\begin{aligned}
G(\omega; \tau) &= H(\omega) - H(\omega|\tau) \qquad (2)\\
&= \mathbb{E}_{\omega \sim P(\omega), a_p^t \sim f_{\theta|\omega}^{PNT}}[log P(\omega|s_p^t, a_p^t)] + H(\omega)\\
&= \mathbb{E}_{a \sim f_{\theta|\omega}^{PNT}}[D_{KL}(log(P(\omega_p|s_p^t, a_p^t))||log(q_{\zeta|\theta}^{\omega}(s_p^t, a_p^t))) + \mathbb{E}_{\omega \sim P(\omega)} log(q_{\zeta|\theta}^{\omega}(s_p^t, a_p^t))] + H(\omega)\\
&\geq \mathbb{E}_{\omega_p \sim \mathcal{N}(\vec{\mu}_p, \vec{\sigma}_p^2), a \sim f_{\theta|\omega}^{PNT}}[log(q_{\zeta|\theta}^{\omega}(\omega_p|s_p^t, a_p^t))] + H(\omega) = L_G(f_{\theta|\omega}^{PNT}||q_{\zeta|\theta}^{\omega})
\end{aligned}
$$

In our approach, we make use of continuous personalized embeddings which allow for greater
expressivity in the embedding space, $\Omega$. As such, we utilize a mean-squared error (MSE) loss
between a sample from the approximate posterior (modeled as a normal distribution with constant
variance) and the current embedding.

We present the approximate normal distribution, $\mathcal{N}_{q_{\zeta|\theta}^{\omega}}$, in Equation 3, where $\omega$ is the mean outputted
by the posterior network, and $\sigma$ is the standard deviation.

$$
\mathcal{N}_{q_{\zeta|\theta}^{\omega}} = \frac{1}{\sigma\sqrt{2\pi}} e^{-\frac{1}{2}\frac{(x-\omega)^2}{\sigma^2}} \qquad (3)
$$

**Theorem 4.1.** *Minimizing the mean-squared error between a sample from the approximate posterior
and the current embedding is equivalent to maximizing the log-likelihood and therefore, the evidence
lower bound.*

*Proof.* The mean-squared error (MSE) loss is $(x - \omega)^2$, where $\omega$ is the sample from the approximate
posterior, and $x$ is the current personalized embedding used to generate the predicted action. This is
equivalent to the exponent numerator in $\mathcal{N}_{q_{\zeta|\theta}^{\omega}}$. With constant variance, the exponential function is

monotonic, and thus, minimizing the exponent will maximize the likelihood of the posterior. Thus, minimizing the MSE is equivalent to maximizing the likelihood of the posterior. This naturally extends to the multivariate case. $\qquad\square$

## 5 Interpretability User Study Details

Here, we present the details of our novel user study to assess the interpretability of our discretized PNTs. We design an online questionnaire that asks users to predict a task to schedule given an input using a decision-tree based method and a neural-network-based method. Each user is randomly assigned a reasoning level, standard, pointwise, or counterfactual. Standard and counterfactual reasoning are discussed in the main paper. Pointwise reasoning outputs a probability of taking a certain action given a feature vector describing that action, $x_a^t$ from state $s^t$, and a contextual feature vector capturing features common to all actions $\bar{x}^t$. We can generate pointwise features through Equation 4.

$$z^{t,p} := [\omega_p, \bar{x}^t, x_a^t], y_a^t = 1 \tag{4}$$

$$z^{t,p} := [\omega_p, \bar{x}^t, x_{a'}^t], y_{a'}^t = 0 \tag{5}$$

The tree and neural network-based models were trained under minimal sizes that were capable of achieving near-perfect accuracy. Tree models are learned PNTs, which are then discretized. The NN models are generated by appending personalized embeddings to a NN and following the training methodology described in Algorithm 1 from the main paper. Then, comparison weights and model weights for the discrete trees and neural networks, respectively, were rounded to the nearest 0.25. Rounding yielded $\sim 2\%$ loss in accuracy but allowed for the survey to be conducted within a reasonable time. For each type of decision-making framework, we provide instructions for how to utilize the framework to make a prediction. The order in which the user completes the neural network portion and decision tree portion is randomized. We explore additional hypothesis: counterfactual tree-based decision-making models are more interpretable (**H4**), quicker to validate (**H5**), and are more easily utilized (**H6**) than neural-network based models of any reasoning level. We then provide further comparisons between tree-based methods of different levels of reasoning.

We use four metrics throughout our user study: interpretability of the decision-making model, interpretability of the overall decision-making process, time to compute an output, and correctness. To verify **H4-H6**, we must compare the counterfactual discretized PNT to a standard neural network, pointwise neural network, and pairwise neural network. As the first case is shown in the paper (standard neural network vs. discretized PNT), we provide the results for the other two scenarios here.

## 6 Survey Results

Our IRB-approved anonymous survey was sent out to adult university students. We collected 35 responses, 14 of standard, 11 of pointwise, and 15 of counterfactual. We filter out responses that put in nonsensical answers (i.e., letters where numbers should be and repeated answers).

**H4:** In comparing a NN with pointwise reasoning to a discretized PNT, we test for normality and homoscedasticity and do not reject the null hypothesis in either case, using Shapiro-Wilk ($p > 0.9$ and $p > 0.3$) and Levene's Test ($p > 0.2$ and $p > 0.3$). We perform a paired t-test and find that counterfactual tree-based models were rated statistically significantly higher than pointwise neural networks on users' Likert scale ratings for model interpretability and overall process interpretability ($p < 0.05$ and $p < 0.01$). In comparing a NN with pairwise reasoning to a discretized PNT, we test for normality and homoscedasticity and do not reject the null hypothesis in either case, using Shapiro-Wilk ($p > 0.1$ and $p > 0.1$) and Levene's Test ($p > 0.4$ and $p > 0.4$). We perform a paired t-test and find that counterfactual tree-based models were rated statistically significantly higher than pointwise neural networks on users' Likert scale ratings for model interpretability and overall process interpretability ($p < 0.01$ and $p < .05$). These results support **H4**.

**H5:** In comparing a NN with pointwise reasoning to a discretized PNT, we perform a Wilcoxon signed-rank test on the per-model time to determine an output and find that tree-based models were not statistically significantly quicker to validate than neural networks ($p = 0.37$). In comparing a

NN with pairwise reasoning to a discretized PNT, we perform a Wilcoxon signed-rank test on the per-model time to determine an output and find that tree-based models were statistically significantly quicker to validate than neural networks ($p = 0.001$). This result provides partial support **H5**.

**H6:** In comparing a NN with pointwise reasoning to a discretized PNT, we perform a Wilcoxon signed-rank test on the per-model time to determine an output and find that tree-based models were statistically significantly achieved higher overall correctness scores compared to NN based models ($p < 0.05$), supporting **H6**. In comparing a NN with pairwise reasoning to a discretized PNT, we test for normality and homoscedasticity and do not reject the null hypothesis in either case, using Shapiro-Wilk ($p > 0.05$) and Levene's Test ($p > 0.2$). We perform a paired t-test and find that users using tree-based models statistically significantly achieved higher overall correctness scores compared to NN based models ($p < 0.001$), supporting **H6**.

# 7 Hyperparameters and Architecture Details

We compare our personalized apprenticeship scheduling approach to several baselines [2, 4, 6, 7, 10, 11]. Throughout this section, we will discuss the architecture, implementation details, and learning rates for all baselines and our algorithm in each domain. The runtime mentioned is in respect to a desktop with a NVIDIA RTX 2080Ti GPU and an Intel i7 processor.

## 7.1 Synthetic Low-Dimensional Environment

Each apprenticeship learning algorithm below is given 50 schedules to learn from and tests on a set of 50 unseen demonstrations.

- For the method of Sammut et al. [10], we utilize an multi-layer perceptron (MLP) with 3 linear layers connected by ReLU activation functions. After the last linear layer, we utilize a log softmax function to compute the log probability of which task to schedule. Each linear layer has 10 hidden units. We utilize the Adam optimizer with a learning rate of $1e^-3$. The runtime for training and verifying this model is under 30 minutes.

- For the method of Nikolaidis et al. [7], we utilize k-means clustering to separate the data into two clusters. Two neural networks (one for each cluster) are trained to imitate demonstrator data within the cluster. Each network utilizes the same architecture and learning rate used in the baseline of Sammut et al. [10]. The runtime for training and verifying this model is under 30 minutes.

- For the method of Li et al. [6], we utilize an simulator-free version of infoGAIL. The policy, discriminator, and approximate posterior are modeled by MLPs with 2 linear layers (32 hidden units) connected by a ReLU activation function, and an output activation function of a softmax, sigmoid, and softmax respectively. We initialize the number of discrete modes to 2. We utilize learning rates of $1e^-4, 1e^-3, 1e^-4$ respectively. For the hyperparameters of infoGAIL, we initialize $\lambda_1$ to 1, $\gamma$ to 0.95, and $\lambda_2$ to 0. The runtime for training and verifying this model is under 30 minutes.

- For the method of Tamar et al. [11], we utilize a neural network with 3 linear layers (10, 2, 2 hidden units, respectively) connected by ReLU activation functions. We use N=5 samples as our hyperparameter to estimate the intention probability distribution $\mathcal{P}(z)$. We utilize a learning rate of $1e^-3$ alongside Stochastic Gradient Descent (SGD). The runtime for training and verifying this model is under 30 minutes.

- For the method of Hsiao et al. [4], we utilize a bidirectional LSTM with attention followed by a linear layer as specified in their paper. For the decoder, we utilize three linear layers connected by ReLU activation functions. We utilize a learning rate of $1e^-3$ alongside Stochastic Gradient Descent (SGD). The runtime for training and verifying this model is under 30 minutes.

- For the method of Gombolay et al. [2], we utilize a standard decision tree (counterfactuals are not possible when $|A| \leq 2$) of depth 10. The runtime for training and verifying this model is under 30 minutes.

- For our Personalized Neural Trees, we utilize a max depth of 6 (32 leaves) and embedding dimension of 2 ($d = 2$). We set learning rates of $\theta$ to $1e^-3$, $\omega$ to $1e^-2$, and $\zeta$ to $1e^-3$.

We find empirically that setting the learning rate of $\omega$ slightly higher allows for better LfD accuracy. For our approximate posterior, $q^{\omega}_{\zeta|\theta}$, we set the value of $\sigma_p$ to zero. The runtime for training and verifying this model is under 30 minutes.

## 7.2 Synthetic Scheduling Environment

Each apprenticeship learning algorithm below is given 150 schedules to learn from and tests on a set of 100 unseen demonstrators.

- For the method of Sammut et al. [10], we utilize an multi-layer perceptron (MLP) with six linear layers connected by ReLU activation functions. After the last linear layer, we utilize a log softmax function to compute the log probability of which task to schedule. Each linear layers have 128, 128, 32, 32, 32, and 20 hidden units, respectively. We utilize the Adam optimizer with a learning rate of $1e-4$. The runtime for training and verifying this model is approximately 30 minutes.

- For the method of Nikolaidis et al. [7], we utilize k-means clustering to separate the data into three clusters. Three neural networks (one for each cluster) are trained to imitate demonstrator data within the cluster. Each network utilizes the same architecture and learning rate used in the baseline of Sammut et al. [10]. The runtime for training and verifying this model is approximately 30 minutes.

- For the method of Li et al. [6], we again utilize a simulator-free version of infoGAIL. The policy follows the same network structure used in the Sammut et al. [10] baseline. The discriminator and approximate posterior are modeled by MLPs with six linear layers (128, 128, 128, 32, 32, 32 hidden units, respectively) connected by a ReLU activation function, and an output activation function of a sigmoid, and softmax respectively. We initialize the number of discrete modes to 3. We utilize learning rates of $1e-4, 1e-3, 1e-4$ respectively. For the hyperparameters of infoGAIL, we initialize $\lambda_1$ to 1, $\gamma$ to 0.95, and $\lambda_2$ to 0. The runtime for training and verifying this model is approximately 24-48 hours.

- For the method of Tamar et al. [11], we utilize a neural network with 5 linear layers (128, 32, 32, 32, 32, 20, 2, 2 hidden units, respectively) connected by ReLU activation functions. We use N=5 samples as our hyperparameter to estimate the intention probability distribution $\mathcal{P}(z)$. We utilize a learning rate of $1e-3$ alongside Stochastic Gradient Descent (SGD). The runtime for training and verifying this model is approximately 3 hours.

- For the method of Hsiao et al. [4], we utilize a bidirectional LSTM with attention followed by a linear layer as specified in their paper. For the decoder, we utilize six linear layers connected by ReLU activation functions. We utilize a learning rate of $1e-3$ alongside Stochastic Gradient Descent (SGD). The runtime for training and verifying this model is approximately 3 hours.

- For the method of Gombolay et al. [2], we utilize a pairwise decision tree of depth 10. The counterfactuals are set to one-hot encodings of each action, as done in the original paper. The runtime for generating and verifying this model is approximately 5 minutes.

- For our Personalized Neural Trees, we utilize a max depth of six (32 leaves) and embedding dimension of 3 ($d = 3$). We set learning rates of $\theta$ to $1e-2$, $\omega$ to $1e-2$, and $\zeta$ to $1e-2$. We find empirically that pretraining the policy network first and then adding in the posterior at a later epoch results in both good performance and mutual information maximization. This is opposed to training both models at once from scratch. For our approximate posterior, $q^{\omega}_{\zeta|\theta}$, we set the value of $\sigma_p$ to zero. The runtime for training and verifying this model is approximately 24 hours.

## 7.3 Taxi Domain

Each apprenticeship learning algorithm below is given 25 successful trajectories from each user and tested on a set of 25 unseen trajectories from each demonstrator.

- For the method of Sammut et al. [10], we utilize the same architecture and learning rate as that of the synthetic scheduling environment. The runtime for training and verifying this model is approximately 30 minutes.

(a) Low-dim Environment                 (b) Survey Environment (counterfactual)

Figure 2: This figure depicts the learned PNT model after translation to an interpretable form.

- For the method of Nikolaidis et al. [7], we utilize k-means clustering to separate the data into three clusters. Three neural networks (one for each cluster) are trained to imitate demonstrator data within the cluster. Each network utilizes the same architecture and learning rate used in the baseline of Sammut et al. [10]. The runtime for training and verifying this model is approximately 30 minutes.

- For the method of Li et al. [6], we utilize the same architecture and learning rate as that of the synthetic scheduling environment. The runtime for training and verifying this model is approximately 24-48 hours.

- For the method of Tamar et al. [11], we utilize the same architecture and learning rate as that of the synthetic scheduling environment. The runtime for training and verifying this model is approximately 3 hours.

- For the method of Hsiao et al. [4], we utilize the same architecture and learning rate as that of the synthetic scheduling environment. The runtime for training and verifying this model is approximately 3 hours.

- For the method of Gombolay et al. [2], we utilize a pairwise decision tree of depth 13. The counterfactuals are set to one-hot encodings of each action, as done in the original paper. The runtime for generating and verifying this model is approximately 5 minutes.

- For our Personalized Neural Trees, we utilize a max depth of 8 (128 leaves) and embedding dimension of 3 ($d = 3$). As counterfactual task information is not readily available, we utilize one-hot encodings for each action. We set learning rates of $\theta$ to $1e{-}2$, $\omega$ to $1e{-}1$, and $\zeta$ to $1e{-}2$. We find empirically that pretraining the policy network first and then adding in the posterior at a later epoch results in both good performance and mutual information maximization. For our approximate posterior, $q^{\omega}_{\zeta|\theta}$, we set the value of $\sigma_p$ to zero. The runtime for training and verifying this model is approximately 12 hours.

# 8   Interpretable Models

As machine learning is being increasingly deployed into the real world, interpretability is required for these systems to gain human trust [1, 3, 8]. Interpretability refers to attempts that help the user understand why a machine learning model behaves the way it does. A clear visualization of a policy is one way to help a human form an accurate representation of its capabilities [9]. Furthermore, an interpretable model of resource allocation or planning tasks would be highly useful for a variety of reasons, from decision explanations to training purposes. In Figure 2, we display interpretable models generated through discretization for the low-dimensional environment and survey scheduling environment.

# 9   Future Work

During the deployment of a discretized PNT, we required pre-inferred embeddings to understand decision-maker behavior. As this involves a sample of the decision-maker's data and the use of backpropagation with a pre-discretized PNT to infer demonstrator style, we feel this can be improved by producing the demonstrator embedding through the means of our approximate posterior $q^{\omega}_{\zeta|\theta}$, modeled as a $PNT \setminus \omega$. This can be discretized following the framework of Section 4.3 of our paper, producing an interpretable model that predicts a mean and covariance of an embedding given a single state-action pair. This discretized posterior then takes in a state-action pair and produce the latent

embedding that generated this action. In this way, the interpretable discretized PNT has a method to naturally infer the demonstrator's embedding.