[Reviews · NeurIPS 2020]

Review 1

Summary and Contributions: An apprenticeship learning model for resource scheduling called Personalized Neural Trees (PNT) The proposed model is based on differentiable decision trees. They introduce a personalized embedding representing person-specific modality. 1. Given the state s and personalized embedding w, the PNT can output the probability of actions as in fig.(1a). 2. Given the predicted action and state, the personalized embedding can be inferred from a posterior model. They aim to maximize the mutual information between embedding and trajectories. In practice, they use the MSE loss between the current embedding and a sample from the approximate posterior. 3. The second loss term is the cross entropy loss between predicted action and the true action 4. Given a new demonstrator, the model can update the embedding space without changing other parameters. The experiments show strong empirical results in two synthetic data and one real-world data set.

Strengths: Significance: The proposed method outperforms existing apprenticeship learning methods on synthetic and real domains. In addition, a feature selection and discritization scheme produces interpretable decision trees that are useful. This is confirmed by a user study. Novelty: Existing DDT methods are extended in various ways. Through introducing an embedding for each demonstrator, the model can account for different demonstrators while still sharing statistical strength between the demonstrators during training (as the rest of the model is shared between demonstrators). The statistical efficiency is further improved by a type of data augmentation scheme based on counterfactual reasoning. Other ideas include the derivation of a VI method to learn the embeddings and a way to discretize the learned models into decision trees. Writing: The paper is excellently written and structured.

Weaknesses: no obvious weaknesses.

Correctness: The method seems correct. The details of the user study are in the appendix and I did not check them in detail.

Clarity: Yes, the paper is clearly written and I appreciated the clear structure of the paper.

Relation to Prior Work: Yes. The previous work is discussed well. Many of the ideas in this paper are drawn from previous work but the authors acknowledge this sufficiently.

Reproducibility: Yes

Additional Feedback:


Review 2

Summary and Contributions: This paper studies the problem of imitation or apprenticeship learning from a variety of demonstrators, where we would like to cover all the modes in the demonstration data, learning a *personalized* apprentice for each demonstrator. In addition, we would like the resulting policy to be interpretable to a human. For interpretability, the authors would like to convert the learned model into a discrete decision tree, and so they use a differentiable decision tree architecture and introduce a method to “discretize” it. For personalization, they learn separate embedding vectors for each demonstrator, so that by conditioning on the embedding we can get a policy specific to a given demonstrator. The authors show with an empirical evaluation that the method is much better at covering the modes of demonstration data, and similarly show with a user study that the interpretable decision tree produced is easier to use.

Strengths: I appreciate the grounding of the paper in producing models that can actually be used for decision making, focusing on interpretability and personalization rather than the typical sole metric of accuracy, and that the resulting methods work well with DAgger training, which is often necessary for better generalization. The proposed architecture and discretization procedure are simple and natural ideas to try, and seem to work well empirically.

Weaknesses: A key goal of this method is to produce personalized policies, but the empirical evaluation does not demonstrate this -- it simply reports accuracy, without explaining whether or how the policy had to condition on the demonstrator identity in order to get that accuracy. The environments in the empirical evaluation seem particularly suited to conditional, hierarchical reasoning, and so the decision tree architecture provides a very useful inductive bias that baselines do not have. I would recommend the authors compare to vanilla decision trees in order to control for this inductive bias.

Correctness: To my knowledge, yes.

Clarity: Yes.

Relation to Prior Work: I believe so, but I am not an expert in this field.

Reproducibility: Yes

Additional Feedback: After reading the author response and other reviews, I am raising my score to 7, primarily because the authors convinced me that decision trees alone are insufficient to solve their evaluation settings. (In fact, this information was in the original paper, but I failed to see it; I apologize for the oversight.) The authors misunderstood my second question (not their fault, it was ambiguous): I was not asking formally how the policies were allowed to condition on the demonstrator identity, which is clear from the mathematical description in the paper. I was asking, *for the specific evaluations the authors ran*, how do the resulting policies depend on demonstrator identity. A potential answer could be “in the Taxi domain, when conditioning on demonstrator 1 the policy will first go to X and then Y, but when conditioning on demonstrator 2 the order will be reversed”. I think this sort of qualitative analysis would improve the paper and encourage the authors to include it in a subsequent version. ---- Questions: How do vanilla decision trees perform in the chosen environments? (The decision trees should be constructed to predict the action given the state and demonstrator identity.) Do the discretized, interpretable policies depend on the demonstrator identity? If so, how?


Review 3

Summary and Contributions: taking into account differences in the behaviors of different human demonstrators. Specifically, they present a variation of differentiable decision trees that learn a set of latent features associated with each human demonstrator, which allows the learned model to capture the differences between demonstrators, while still learning from the similarities in their behavior. When learning from a new demonstrator, the algorithm can infer these latent parameters online as it observes more data. They also incorporate a mechanism that allows the continuous DDT tree to be discretized, making the model more easily interpretable by a human user. The algorithm is evaluated within the apprenticeship learning for scheduling framework described in [11] "Apprenticeship Scheduling: Learning to Schedule from Human Experts" (Gombolay et al. 2016), where scheduling is treated as a sequential decision problem, and decision trees are used to learn policies for this problem from expert demonstrations (in the form of schedules). Experiments with both synthetic and human generated data (where data is heterogeneous between demonstrators), show that in this setting the PNT algorithm outperforms a number of existing imitation learning baselines (including [11]) in the scheduling domain. The paper evaluates the interpretability of the decision trees learned by PNT through a small scale user study in which participants are asked to apply either the learned decision tree, or a small neural network, to classify a set of examples, computing the outputs of the models by hand. This study shows, unsurprisingly, that participants were faster and more accurate in computing the outputs of the decision trees than those of the neural networks.

Strengths: The main strength of this work is the novel PNT algorithm for learning interpretable decision trees from heterogeneous human demonstrations. The authors demonstrate that PNT is able to learn accurate scheduling models from heterogeneous data in tasks where all existing methods fail. They further show that PNT does not sacrifice the ability of decision-tree algorithms to produce human-readable models, by learning models that can be easily discretized once trained. Interestingly, the discretization process used to generate interpretable models also seems to improve prediction accuracy in some tasks, which may be due to the fact that the interpretable model is less complex than the original model, preventing overfitting and improving generalization.

Weaknesses: A major point of concern is the value of the user study in comparing the interpretability of the scheduling decision tree to that of a neural network. While there is nothing wrong with the study methodology itself, or subsequent statistical analysis, it is not clear that the user study is really capturing the difference in interpretability between the neural network and the decision tree. IT seems that the study may not really be testing the hypotheses (H1, H2, H3) that the authors claim it is. The study asks participants to manually compute the outputs of decision trees and neural networks, given their decision thresholds (for trees) or weights and biases (for networks). It is not surprising that it is easier to manually execute a decision tree, as many fewer operations are needed, but this question seems spurious. In practice, we would never expect a user to manually compute the output of a neural network (or a large decision tree for that matter). The relevant question for "ease of use" (H2) might better be phrased as "does access to the structure of the decision tree make it easier to use in a particular application versus a black-box model (tree or network)?". We could ask whether knowing the rules that led to a decision would give a user more confidence in the outcome when it conflicts with their own judgement, or allows them to detect cases where it may have made an error (or incorrect inputs were give). Similarly, "ease of validation" (H3) would be better phrased as "how easy is it to debug this model?" Given a flawed model, we could ask how likely a user is to detect the flaws (based on observed input-classification pairs, and access to the model structure), and how easy it is for them to repair those flaws. For H1, the notion of "interpretability" referred to in the questionnaire seems vaguely defined. We can't really know what a participant would think "interpretable" means in this context, even if they are experts in machine learning. All we can really learn from the questionnaire is that the task of manually executing the decision tree was easier, which is unsurprising, but also tells us little about how usable either model is in practice. It is also unclear what new insights the study presented in this work provides over similar previous studies in "Optimization Methods for Interpretable Differentiable Decision Trees in Reinforcement Learning" (Silva et al. 2020). The novelty of the work is somewhat limited, as it is largely an extension of [11] "Apprenticeship Scheduling: Learning to Schedule from Human Experts", to the case of heterogeneous experts. This extension is itself non-trivial, however, as it requires a novel variational algorithm for learning decision trees with latent, demonstrator-specific variables. It would be helpful, however, to include an ablation in which the base DDT algorithm is used to learn the decision trees for scheduling, to illustrate the importance of explicitly modelling the demonstrator-specific aspects of the data. It would also be helpful to include more detailed discussion of how the demonstration data for the synthetic scheduling task was generated, and along what dimensions demonstrations for this task varied between demonstrators. This is covered in the related work, but given the importance of personalization to the contribution of this paper, it is worth discussing it in the main document. Finally, the task performance presented in Table 1 appears to be in terms of scheduling choice accuracy. It would be helpful to include information on the relative qualities of the schedules resulting from these learned models, if such data is available.

Correctness: The algorithm presented in this work appears to be sound, and no apparent flaws in the methodology used to evaluate the new algorithm. The user study is technically sound in how was conducted, and how its results were analysed, but it is unclear that the study adequately addresses the hypotheses that the authors intended.

Clarity: Overall the paper is well-written and clearly describes the new method, as well as the experimental setups and results for both performance evaluations and the results of the user studies. The paper suffers somewhat from attempting to include a large amount of content in limited space. This has lead the authors to push some important content to the supplementary materials, including the synthetic data generation procedures, and most details of the user study. The authors may want to reconsider what details they include in the main document, and which they leave for the appendix (e.g. moving exhaustive discussions of statistical tests and significance values to the appendix).

Relation to Prior Work: The paper discusses a large body of related work on the problem of learning from multiple human demonstrators with heterogeneous behaviors. It makes clear that the goal of this work is to learn "interpretable" models from heterogeneous demonstrations, in the form of decision trees. Most importantly, the work compares the proposed method against several of these baselines in all of the experimental tasks, demonstrating that the new method significantly outperforms these methods in these particular tasks. At the same time, the paper acknowledges that many of the baseline methods were not primarily developed for the types of scheduling tasks considered in this work, and includes a stronger baseline [11] that is well suited to these types of tasks. There does appear to be an error in an important bibliography entry. [11] is titled "Decision-Making, Authority, Team Efficiency and Human Worker Satisfaction in Mixed Human-Robot Teams", but the reference appears to refer to "Apprenticeship Scheduling: Learning to Schedule from Human Experts".

Reproducibility: Yes

Additional Feedback: POST REBUTTAL: While I agree that the study design is robust against varying interpretations of "interpretability", what this really means is that the study can accurately tell us how a random person drawn from the participant population would respond when asked whether the method is "interpretable". This does not tell us what "interpretability" means, or whether the participants' understanding of the term matches our intuitive definition. That said, agreeing on a universal definition of interpretability for machine learning is beyond the scope of this work. It would be helpful, however, to include at least a couple of sentences explaining that the objective elements of the study (e.g. the accuracy of the users' calculations), measure a very specific definition of interpretability that may be of limited practical use. The authors could also clarify what their working definition of "interpretability" is, and highlight that there is no way to know from this study whether participants us a similar definition. The question of what makes a model "interpretable" in a particular context is difficult to answer, but of critical importance. While the user studies presented are not convincing as to the superior interpretability (and more importantly, user friendliness) of decision trees, there are likely many scenarios in which this hypothesis is true. The authors may want to carefully consider how "interpretability" can be most usefully defined, and what human-subjects experiments would be best suited to test this definition.


Review 4

Summary and Contributions: This paper presents an approach to personalized learning from demonstration approach focused on the case where different users demonstrate heterogeneous policies (i.e., each user has a different style of policy). Their approach has two components: 1) in order to personalize they condition their model on a learnable user embedding; 2) in order to make their model interpretable, they use a extension of differentiable decision trees that can be converted to crisp decision trees later on.

Strengths: - The interpretable formulation of differentiable decision trees, and seeing how they do not lose much performance when converted to crisp trees is quite interesting. - Going the extra mile to integrate PNTs with DAgger was nice to see!

Weaknesses: - I found this paper was trying to pack too much into a single paper: the personalization and the interpretability works are totally orthogonal, and I'd say should have gone into separate papers. - Although this is a minor weakness, the evaluation domains used are fairly simple. Even the Taxi domain that is labeled as "real world", is a toy grid world. So, it's unclear how do the PNT approach scale to actual real world domains. [edit after rebuttal: I just wanted to clarify, that this is more of a criticism on the language used in the paper, than on the experiments per se. I think experiments in small domains are fine, but please do not call them "real world".]

Correctness: - Results seemed correct to me

Clarity: - The paper is well written and was mostly easy to follow. Perhaps some of the technical parts could use more intuitions before the equations, but it was mostly understandable.

Relation to Prior Work: - Relevant related work was sufficiently covered, I just had one suggestion (see comments below), but other than that I thought literature was covered.

Reproducibility: Yes

Additional Feedback: - abstract: "Further, a user study conducted shows that our methodology produces both interpretable and highly usable models (p < 0.05)." -> This sentence needs revising, it does not make sense to attach a p value to such a vague statement. So, please rephrase with the exact hypotheses that this p value is attached to (specifically "more interpretable than neural network models") - page 1: "approaches assuming homogeneous demonstrations either fit the mean (i.e., driving straight into the car ahead of you) or fit a single mode (i.e., only pass to the left)." -> this is not true in general. See for example "A Dynamic-Bayesian Network framework for modeling and evaluating learning from observation" (2014), where this problem is addressed explicitly. The general problem is not just "demonstration heterogeneity", but modeling the action probability distribution, rather than the action that minimizes prediction error. Even if all demonstrations came from the same user, if such user randomly passed on the left or right, the same problem would arise. - Question on page 4 concerning PNT: how is the structure of the tree determined? (how many layers/leafs/etc.) in standard decision trees this is learned from data, but I am assuming here it is given beforehand, right? [edit after rebuttal: thanks for the clarification!] - page 4: "we observe the decision, a, that person, p, made at time, t" -> " we observe the decision, a, that person p made at time t" - pages 4/5: concerning the counterfactuals. Equations 3-4 seem to indicate that ALL other actions are used as counterfactuals. This seems ok for deterministic policies, but can that be guaranteed? if policies are stochastic, it could be that the user chose one action, but could have perfectly picked some other action as well. Did you encounter this problem? I wonder if sampling a small set of other actions, rather than using all of them would be a better approach in this case. [edit after rebuttal: thanks for the clarification!] - page 5: "each leaf node dictates a single action to be taken" -> in standard decision trees, it is common that leaves can actually have a mix of labels. This is useful, again for stochastic policies. Did you experiment with this? Specially, when the argmax action in a leaf is only marginally more likely than the second highest, this should help while not getting too much in the way of interpretability. [edit after rebuttal: thanks for the clarification!] - page 5/6: the training procedure was not clear to me: the "online" procedure is said to be run for "new human demonstrators". However, during the "offline" procedure all demonstrators are also new, right? So, I do not see why the "online" case is special and the tree weights have to be frozen. This is basically like the "pre-training/fine-tuning" standard process. Did you observe issues if \theta wasn't frozen? [edit after rebuttal: thanks for the additional explanation. I have not checked supplementary material in detail, but in case the explanation in the rebuttal is not included, please do so in case of acceptance!] - page 6: "Real-world Data: Taxi Domain" -> please do not call the synthetic toy Taxi domain "real-world data". - page 7: "which is acknowledge to" -> "which is acknowledged to" - page 7: "InfoGAIL ([20])" -> "InfoGAIL [20]" - page 8: For the user study, what was the size of the PNT trees given to them? Tree models are more interpretable than neural networks as long as they are kept small (in the same way that small neural networks, e.g. simple linear regression is also interpretable), but grow unmanageable with size.

[Author Response · NeurIPS 2020]

**[R2]** *Explanation of how policy conditions upon the demonstrator identity during empirical evaluation* – Our method
starts by setting the personalized embedding, $\omega_{p'}$, for a new human demonstrator, $p'$, to be the mean of the embeddings of
demonstrators in the training set. After every timestep, we update the personalized embedding utilizing the information
provided (i.e., the true action) to converge on the type of current demonstrator in embedding space.

**[R2]** *Decision tree architecture provides inductive bias* – In Table 1, we report the results for a vanilla decision tree
with the only modification being that we also provide the PNT's embeddings as additional inputs to the DT to help
the DT tease out the heterogeneity. Unsurprisingly, this approach performed poorly. If we train a vanilla DT without
embeddings or pairwise comparisons, the accuracy is only $55.76 \pm 1.4\%$, $32.4 \pm 0.7\%$, and $74.9 \pm 0.2\%$ across the
three domains, which is significantly worse than our PNT approach. We believe these comparisons provide a control for
the inductive bias in question and demonstrate the advantage of our formulation.

**[R2]** *Interpretable policy's dependence on demonstrator identity* – Discretized PNTs learn splitting criterion dependent
on the demonstrator embedding, $\omega$. Please see Figure 2 in the supplementary for a depiction.

**[R3]** *Novelty and impact of our user study* – We show through our user study that PNTs are perceived as more
interpretable (verified by our Likert survey), less difficult to use (measured by time of completion), and less difficult to
validate (measured by correctness) than a neural network. To the best of our knowledge, there has not been a prior study
that provides insights into the interpretability of PNTs and NNs in the context of a "style" or "personality" variable(s).

**[R3]** *Rephrasing Hypothesis* – We thank R3 for these suggestions. We will modify **H2** as suggested. We propose to
reword **H3** to say that "our approach is easier for a human to simulate than a neural network." as we think that this
phrasing is a closer description of the task we asked participants to complete.

**[R3]** *"Interpretability" is vaguely defined* – We agree that our study left the definition of "interpretability" open to the
interpretation of the participant. However, different interpretations would be a random effect across participants as
we used a within-subjects design. As such, the experiment design is robust to between-participant variations in the
interpretation of this term.

**[R3]** *Results of base DDT Algorithm* – The base DDT achieves only $55.28 \pm 1.8\%$, $52.35 \pm 0.7\%$, and $76.70 \pm 0.7\%$
accuracy in the Low-dim, Scheduling, and Taxi domains, respectively, whereas our PNT achieves $97.30 \pm 0.3\%$,
$96.13 \pm 2.3\%$, and $88.22 \pm 0.6\%$, clearly displaying the advantage of our framework.

**[R3]** *Details of synthetic schedule generation* – This information is located within the supplementary. We will add these
details into the main paper.

**[R3]** *Quality of resultant schedules* – We only have access to the decision-making policy rather than a ground-truth
metric on the quality of a schedule. However, we agree this would be an interesting metric to pursue in future work.

**[R4]** *Personalization and interpretability works are orthogonal* – While we agree that this paper has an abundance of
material, we believe that model personalization and interpretability are synergistic. Our results show that the mechanism
we leverage for interpretability (i.e., DDTs) improves accuracy in conjunction with personalization, and personalization
helps improve the accuracy of the DDT.

**[R4]** *Evaluation domains are simple* – While the Taxi domain may have a smaller state space, the demonstration data
(in the form of decision trees) collected from 80+ users shows that there is an extremely diverse set of possibilities
when it comes to "solving" even the simple problem of pickoff-dropoff. Our scheduling environment provides a more
clear assessment of how our PNTs work in higher dimensional state spaces.

**[R4]** *PNT hyperparameters* – The PNT structure (e.g., depth, personalized embedding cardinality) is determined by
cross-validation on a subset of the training data.

**[R4]** *Counterfactuals for a stochastic policy* – In our supplementary material, we show that our approach is robust to
noisy demonstrations (i.e., ones in which the users' most-preferred action is not always selected).

**[R4]** *Leaf node representation* – Our PNT leaf nodes represent a probability distribution over actions. While we have
only made use of the argmax over this distribution, we propose to explore sampling in future work.

**[R4]** *Freezing $\theta$* – We freeze $\theta$ to 1) avoid overfitting, 2) maintain a high level of mutual information among the
embeddings and trajectories, and 3) maintain interpretability among our discrete trees. If $\theta$ were to be updated, the
posterior network $q^{\omega}_{\zeta|\theta}$ would need to be retrained to maintain the current level of mutual information. Similarly, the
PNT was trained with regularization offline to afford a discrete tree that allows $\omega$ to continue to vary.

**[R4]** *PNT and neural network sizes in user study* – Both the PNT and NN sizes were minimized while maintaining high
accuracy. Further details are provided in the supplementary material.

**[R1]** We are greatful for your time and feedback – thank you.

[Meta-Review · NeurIPS 2020]

This is a clear case; four reviewers recommend acceptance after a discussion about how the author response has provided additional evidence sufficient to clarify most of the unclarities. The authors are requested to implement the changes they have promised in their response.